

# Urban pluvial flood risk assessment - data resolution and spatial scale when developing screening approaches on the micro scale

Roland Löwe[1] and Karsten Arnbjerg-Nielsen[1]

[1]Section of Urban Water Systems, Department of Environmental Engineering, Technical University of Denmark (DTU), Kgs. Lyngby, Denmark.

**Correspondence:** Roland Löwe (rolo@env.dtu.dk)

**Abstract.** Urban development models typically provide simulated building areas in an aggregated form. When using such outputs to parametrize pluvial flood risk simulations in an urban setting, we need to identify ways to characterize imperviousness and flood exposure. We develop data-driven approaches for establishing this link, and we focus on the data resolutions and spatial scales that should be considered. We use regression models linking aggregated building areas to total imperviousness, and models that link aggregated building areas and simulated flood areas to flood damages. The data-resolutions used for training regression models are demonstrated to have a strong impact on identifiability, with too fine data resolutions preventing the identification of the link between building areas and hydrology, and too coarse resolutions leading to uncertain parameter estimates. The optimal data resolution for modelling imperviousness was identified to be 400m in our case study, while an aggregation of the data to at least 1000m resolution is required when modelling flood damages. In addition, regression models for flood damages are more robust when considering building data with coarser resolutions of 200m than for finer resolutions. The results suggest that aggregated building data can be used to derive realistic estimations of flood risk in screening simulations. Future work needs to focus on training regression approaches where different degrees of flood-awareness in landuse management can be considered.

## 1 Introduction

The development of pluvial flood risk adaptation measures in urban areas typically requires that a variety of combinations of different measures are tested (Radhakrishnan et al., 2018; van Berchum et al., 2018). In addition, flood risk is strongly affected by climate change, urbanization and socio-economic changes (Di Baldassarre et al., 2015; Hinkel et al., 2014; Muis et al., 2015; Muller, 2007; Semadeni-Davies et al., 2008). Projections of these parameters are subject to substantial uncertainties over infrastructure lifetimes between 30 and 100 years (Cohen, 2004; Granger and Jeon, 2007; Hall et al., 2014; Madsen et al., 2014).



To consider these uncertainties in the design of water infrastructures, scenario assessments are performed. In these assessments, model simulations of the urban layout are linked to water systems models (Urich and Rauch, 2014) and the combined impact of climate change, represented as changing forcing in the water systems model, and changes in exposure, represented

by varying simulated urban layouts, is assessed. For example, Löwe et al. (2017, 2018) linked a vector-based urban development model to a 1D-2D hydraulic model of the urban catchment to assess pluvial and coastal flood risk, while Mustafa et al. (2018) implemented a similar setup for fluvial flood risk, considering a cellular automata model for urban development and 2D hydraulic simulations. Other studies have applied cellular automata to study the effect of urbanization on extreme rainfall and resulting flood risk (Huong and Pathirana, 2013) and to quantify changes in coastal flood areas as a result of urbanization

(Sekovski et al., 2015).

Raster-based implementations for modeling urban development such as the one used by Mustafa et al. (2018), have the advantage of short simulation times. In combination with flood screening tools, such tools enable testing flood risk adaptation measures in a scenario-based approach, where the combination of various potential measures and different socio-economic and climate scenarios easily leads to simulation requirements exceeding 10,000 events (Kwakkel et al., 2015; Löwe et al., 2017,

2018; van Berchum et al., 2018). In this context, conceptual flood screening tools as described by Bermúdez et al. (2018) and Jamali et al. (2018) may be preferable over machine learning techniques (e.g., Wang et al. (2015)) because they allow for a physically interpretable implementation of surface adaptation measures, and because they can be linked to conceptual models of the drainage system, and thus be used for a combined assessment of flood risk and other environmental impacts of the drainage system.

When applying a linked (possibly conceptual) urban development – hydraulic simulation setup for pluvial flood risk assessment, we need to consider effects of increasing impervious areas, leading to increased runoff and thus larger flood hazard (Kaspersen et al., 2017), as well as of increasing exposure, resulting from an increase in the potentially flood-prone urban area (Löwe et al., 2017). For both parameters, urban development simulations will frequently not provide a full quantification of the hydrologically relevant variables. For example, impervious surfaces such as terraces, carports or even streets might not

be explicitly represented in the urban development model. Similarly, micro-scale flood damage assessments where simulated flood areas are overlaid with building and infrastructure objects are state-of-the-art in urban hydrology (Hammond et al., 2015), but some building types that are relevant for flood damage assessments (e.g., schools) might not be modeled, and the location of buildings may not exactly reflect reality, or may be blurred if (raster-based) cellular automata approaches are applied. In addition, while Bruwier et al. (2018) clearly demonstrated that building data affect urban flood simulations by blocking flow

paths, this effect is difficult to consider if an urban development simulation only provides building information in the form of a building area density.

For the case where urban development models provide aggregated, raster-based outputs, it is not clear how to link this output to hydrological modeling approaches and subsequent economic pluvial risk assessments. Related work has applied ad-hoc definitions (Löwe et al., 2017), guesstimates from planning documents (Bach et al., 2013) and manual tuning of model

parameters (Bach et al., 2018) to predict imperviousness based on modeled building areas.





Data-driven, empirical approaches would be highly attractive to parametrize this link. Our aim is to evaluate such procedures and to characterize the data resolutions and spatial scales for which robust performance can be obtained. Similarly, for damage assessment we would be highly interested in procedures that allow upscaling of locally derived depth-damage functions, which are likely to provide better damage estimates (Cammerer et al., 2013) and facilitate acceptance by stakeholders. This need was also recognized in the literature (de Moel et al., 2015). Upscaling procedures were previously described by Kreibich et al. (2010) and Thieken et al. (2008), but focused on meso-scale damage assessments, rather than assessments on city- or even neighborhood scale that we are interested in when performing exploratory modelling for urban flood adaptation.

None of the previous work has explicitly assessed to what extent data resolutions applied in the development of scaling procedures affect the outcome of these procedures, and at which spatial scale reasonable predictions can be obtained. A thorough assessment of these issues throughout the pluvial urban flood risk modeling chain is the main contribution of this paper.

## 2 Study area and data

We consider the city of Odense, Denmark as a case study. Odense has approximately 200,000 inhabitants and it is located in a typical moraine landscape close to the sea.

As base data characterizing the urban form, we were provided with building footprints in vector format by Odense Municipality (Figure 1). The building footprints included information on the building types that were grouped into the 11 classes shown in Table S1. In addition, information on the number of residential units and the commercial floor space area in each building was available.

Data on impervious area were provided in vector format. The data were obtained from remote sensing campaigns and grouped into six classes (Figure 3). The responsible utility Vandcenter Syd continuously performs manual, small-scale evaluations of which percentage of each impervious area class is connected to the sewer system. These evaluations were performed for each of the 18,000 subcatchments used in the existing hydrodynamic model for the city's drainage network. We used this processed dataset for our analysis, i.e., impervious area was considered as effective impervious area connected to the pipe system. A digital elevation model (DEM) was available from Agency for Data Supply and Efficiency (2017) in a resolution of 0.4m. The data supplier ensured hydrological validity of the data by removing obstacles for major flow paths such as bridges. The data were averaged to a resolution of 5m.

Figure 1 shows terrain elevations, footprints of the existing buildings and the network of existing major roads. We refer to Löwe et al. (2019) for a detailed evaluation of the characteristics of the urban layout in the case study area.

## 3 Methods

Figure 2 illustrates the overall problem. Hydrological modeling and flood damage assessment are commonly performed based on polygon data characterizing the urban layout. Fast urban development models that are useful for exploratory modeling would typically provide outputs resembling those where building areas were rasterized to resolutions between 25 and 500m.

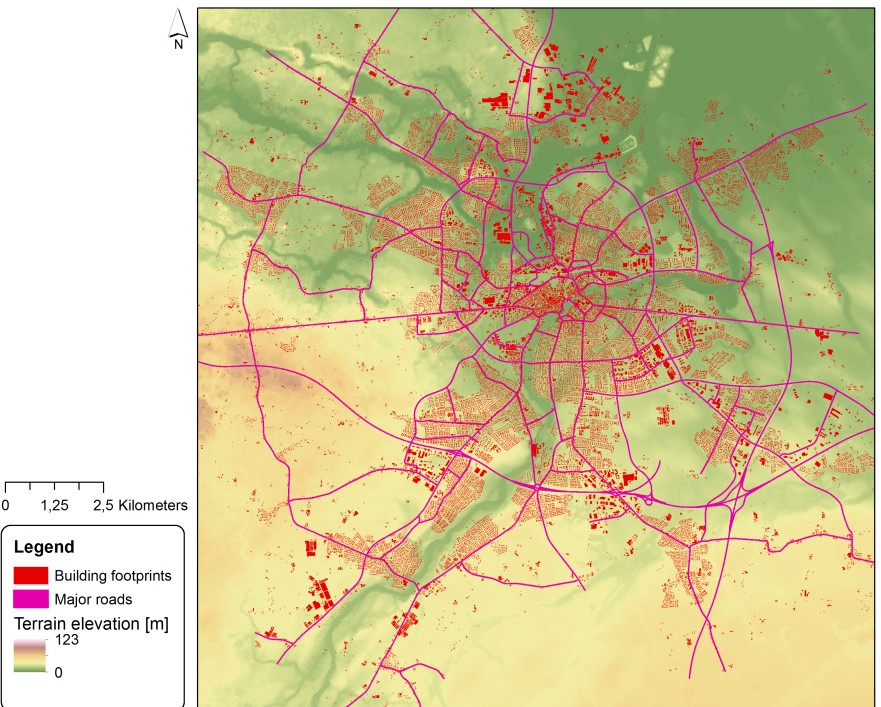

**Figure 1.** Terrain elevations in Odense, footprints of buildings existing in 2017 and major road network.

Such coarse input data will affect both rainfall runoff simulations, i.e., the location where flood hazards occur, and are likely to be incompatible with flood damage assessments derived for polygon data.

To analyze the issues arising in different parts of the pluvial flood risk modeling chain, we have structured our study around steps illustrated in Fig. (3). Summarized roughly, these steps involved the identification of a regression relationship between rasterized building footprint areas (the assumed urban development modeling output) and impervious area. The identified relationship was subsequently applied to derive a raster of predicted impervious area, which was used to parametrize 2D hydrodynamic simulations of surface water flow. The results of these simulations were used to estimate the amount of flooded building area, which was then used as input to regression models that predicted flood damages derived in a reference simulation. The reasoning behind this approach was the following:

1. Urban development models in general, and fast, raster-based modeling approaches in particular, do not provide detailed information on all impervious areas in a catchment. Thus, we need to estimate empirical relationships between an assumed urban development modeling output (here raster-based building footprint areas for different building types) and measured imperviousness. Fitting the regression relationship to datasets with varying resolutions, provides insight on the spatial scale at which the link between urban layout and imperviousness can be identified. Generating predictions at varying resolutions provides insight on the spatial scale at which reasonable predictions can be generated.


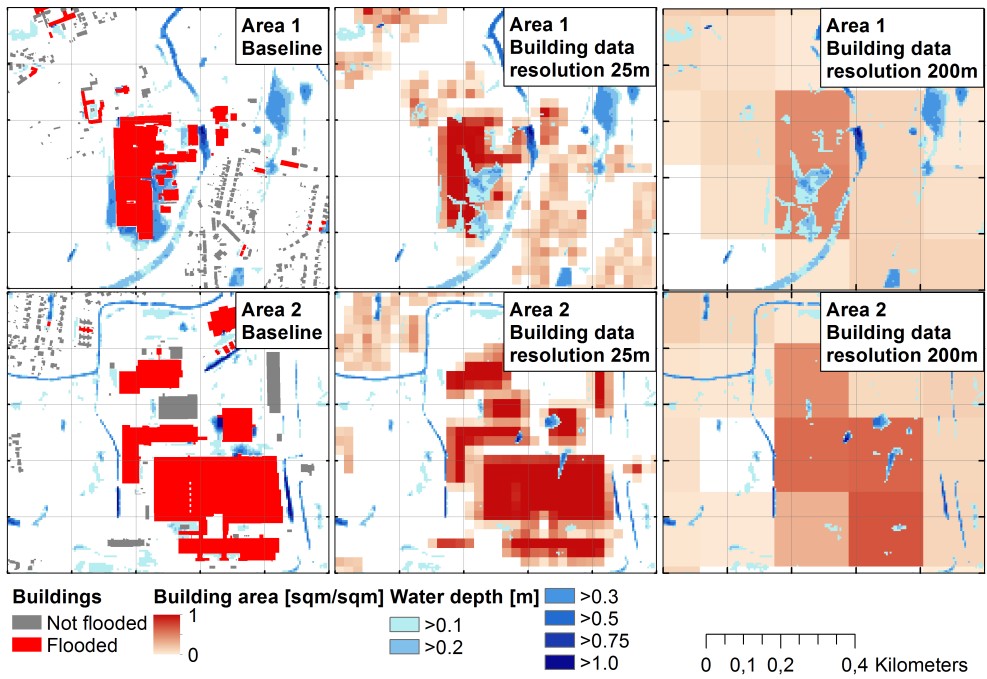

**Figure 2.** Building footprint polygons and total building footprint areas aggregated to data resolutions of 25 and 200m. Shown for two selected areas in the case study together with flood maps simulated based on the building dataset shown in each subfigure for a return period of T=100 years. In the baseline flood simulation, buildings were included as obstacles in the terrain model, while this was not the case in the flood simulations performed for the aggregated building data sets.

2. In a hydrological model, coarse representations of imperviousness affect the amount and location where runoff occurs, and will thus lead to different simulations of flood hazards. We performed hydrodynamic 2D flood simulations where the hydrodynamic model was parametrized using impervious areas based on building areas with varying levels of aggregation. Comparing the resulting flood maps against a reference simulation, we can quantify how increasingly coarse representations of the urban layout affect simulated flood hazard.

3. Economic flood damages are an important parameter in decision making related to flood adaptation. The standard approach for damage estimation in urban hydrology is to overlay high-resolution flood areas and building polygons. If only coarse, raster-based building data are available, flood damages can be derived by establishing a regression relationship between flood damages derived in a reference simulation and the amount of flooded building area as a measure of exposure. Inspecting the validity of this relationship provides insight into the combined impact of coarse representations of the urban layout on hazard and exposure.

In addition to the above, buildings affect simulated flood hazards by obstructing flow paths. This effect cannot be considered when only coarse building data are available. To separate this effect in our study, we performed an additional baseline simula-
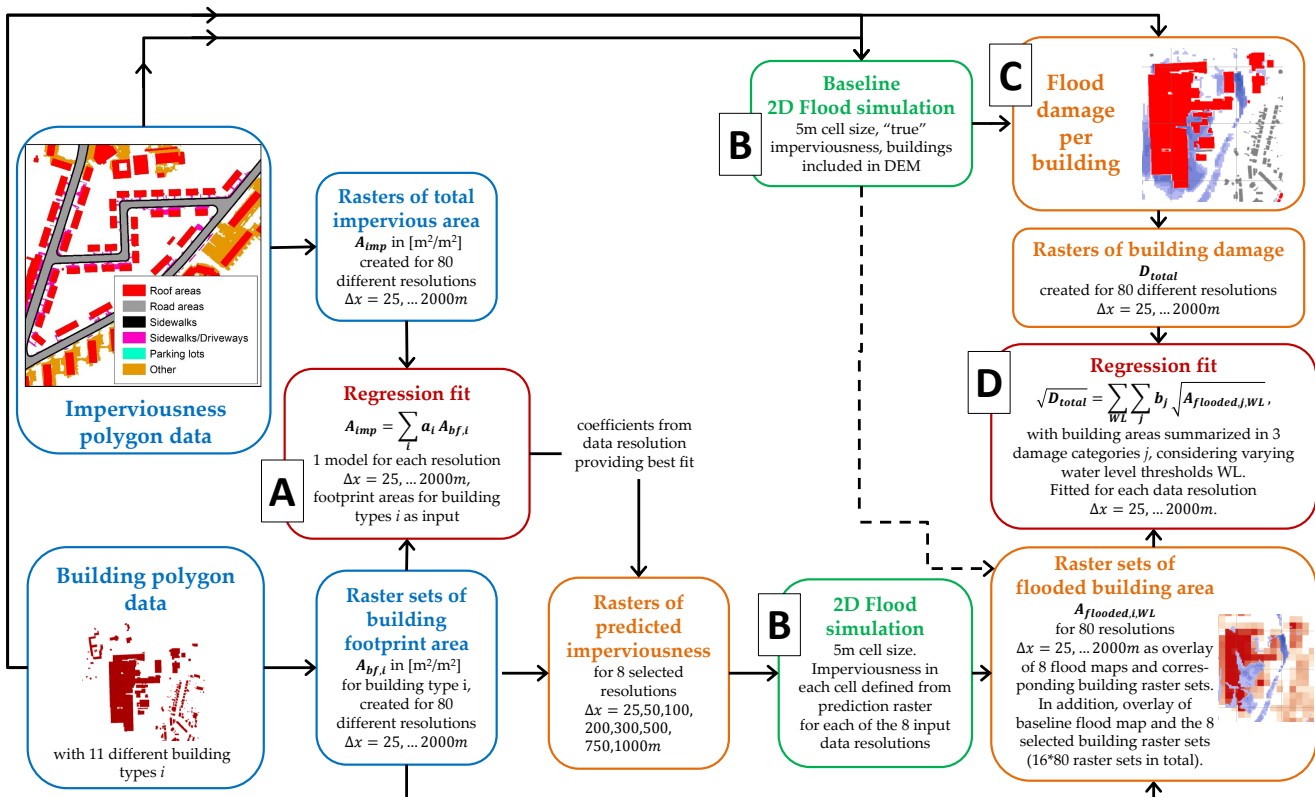

**Figure 3.** Outline of the analysis steps performed in this paper. Letters A to D refer to the part of the methods sections were the corresponding step is detailed. The dashed line illustrates the case where flood maps from the baseline simulation were used to derive flooded building areas as input to damage regression. Note that the second baseline 2D flood simulation where buildings were not inserted in the DEM is not shown in the flow chart.

tion where buildings were not included in the DEM (not shown in Fig. (3)). We compared simulated flood areas and damages against the reference.

## 3.1   A - Regression fitting to predict impervious area

### 3.1.1   Model setup

Our aim was to predict impervious area in simulated urban developments when the assumed output of an urban development
are building footprint areas for different building types. Linear regression approaches for modeling such relationships were previously documented by Butler and Davies (2011) for detached housing only, and by Chabaeva et al. (2009) for a variety of land cover classes derived from satellite observations. To identify a regression relationship, we rasterized the high-resolution





polygon data and, for each pixel $j$, modeled the observed impervious area $A_{imp,j}$ in $m^2/m^2$ against the building footprint area $A_{bf,i,j}$ in $m^2/m^2$ for each of the building types $i$ shown in Table S1. We considered the following relationship:

$$A_{imp,j} = \sum_i a_i \cdot A_{bf,i,j} \tag{1}$$

We have not included an intercept in Eq. (1) to ensure undeveloped areas are assigned an imperviousness of 0, and because an analysis of the dataset suggested no need for an intercept (Figure S1). For fine data resolutions this leads to biased regression predictions. While the dataset certainly is subject to spatial autocorrelation, the regression models provided strong predictive performance and we have therefore not investigated the matter further.

To test the impact of spatial data resolution, we fitted regression models to datasets with 80 different resolutions $\Delta x_{fit}$ ranging from 25 to 2000m in steps of 25m. The regression coefficients identified for each resolution were then used to predict imperviousness at 80 different aggregation levels $\Delta x_{pred}$ ranging from 25 to 2000m. We embedded our tests into a cross-validation setup where 80% of the dataset were used for calibration and 20% for model validation. If $\Delta x_{pred} > \Delta x_{fit}$ we sampled from the pixels of the dataset used for prediction, and otherwise from the pixels of the fitting dataset. For cross-135 validation, a pixel from the dataset with finer resolution was linked to the pixel of the dataset with coarser resolution with which it shared the greatest overlap. The cross-validation procedure was repeated $k = 1000$ times, i.e., a total of $80 \cdot 80 \cdot 1000$ regression models was considered.

### 3.1.2 Performance assessment

During each iteration, we computed bias ratio $RBIAS_{Aimp,k}$, $RMSE_{Aimp,k}$ and $NSE_{Aimp,k}$:

$$RMSE_{Aimp,k} = \sqrt{1/n \cdot \sum_j \overline{(A_{imp,pred,j} - A_{imp,obs,j})^2}} \tag{2}$$

$$NSE_{Aimp,k} = 1 - \frac{\sum_j (A_{imp,pred,j} - A_{imp,obs,j})^2}{\sum_j (A_{obs,pred,j} - \overline{A_{imp,obs}})^2} \tag{3}$$

$$RBIAS_{Aimp,k} = \sum_j A_{imp,pred,j} / \sum_j A_{imp,obs,j}, \tag{4}$$

where $A_{imp,pred,j}$ and $A_{imp,obs,j}$ were predicted and observed impervious areas for a pixel $j$ in the validation dataset and $\overline{A_{imp,obs}}$ was the average imperviousness of all pixels $j$ in the validation dataset. We considered the median of $RBIAS_{Aimp,k}$ 145 and $NSE_{Aimp,k}$ over all $k$ iterations as measures of goodness of fit, and the standard deviation $\sigma(RMSE_k)$ of $RMSE_{Aimp,k}$ as a measure of how reliably the model could be identified for a given combination of $\Delta x_{fit}$ and $\Delta x_{pred}$.

### 3.2 B - 2D flood simulations

### 3.2.1 Model setup

We performed 2D flood simulations of pluvial hazards for ten different models, considering:





– a model where imperviousness was determined from the original imperviousness dataset, and where buildings were
       included in the DEM (baseline model),

       – a model where imperviousness was determined from the original imperviousness dataset, and where buildings were not
       included in the DEM (baseline without buildings), and

       – models where imperviousness was derived considering the regression relationship shown in the supporting material
(Sect. S2), and considering building data aggregated to resolutions of 25, 50, 100, 200, 300, 500, 750 and 1000m as
       input. Buildings were not included in the DEM in this case.

Our 2D modelling approach was the exact same as used by Kaspersen et al. (2017) for the same case study area. The
2D surface flow model was implemented in MIKE 21 (DHI, 2016) using a grid size of 5m. Simulations were performed for
Chicago design storms (CDS) with return periods of 20 and 100 years and durations of 4 hours. Rainfall-runoff computations
were performed for each grid cell during each time step of a simulated event, and the runoff created in each cell was then
included in the simulation of surface water flows.

As in Kaspersen et al. (2017), runoff $R_t$ in time step $t$ for each 5m pixel was computed as

$$R_t = P_t - f_t(1 - IS) - P_{t,RP5}IS, \tag{5}$$

where $P_t$ was the rain intensity and $IS$ the ratio of impervious area in a pixel to its total area. The effective infiltration
intensity $f_t(1-IS)$ in a cell was computed based on a constant infiltration rate $f_t = 29.3 mm \cdot h^{-1}$. On the impervious portions
of a pixel, the rain intensity $P_{t,RP5}$ of a 5 year design storm at the same time step $t$ was subtracted from the rain intensity to
simulate the effect of drainage systems.

Impervious areas linked to major roads (Figure 1) were preserved throughout all simulations. In an urban development
simulation, main roads would need to be considered explicitly, instead of being lumped into a regression prediction of imper-
viousness with building areas as the only input. As an example, we included maps of infiltration rates $f_t(1-IS)$ derived for
two building datasets in the supporting information, Sect. S3.

### 3.2.2   Performance assessment

We compared the simulated flood maps against the baseline simulation where true imperviousness percentages were applied
for runoff modeling, and buildings were included in the DEM. In the comparison, we focused on built-up areas and excluded
natural areas and water bodies.

We created contingency tables where we counted in how many pixels both the predicted flood map under scrutiny and the
baseline flood map exceeded a water level of 0.1m (hits), and how often this was the case only for the baseline model (misses)
or the tested model (false alarms). Subsequently, we computed the scores hit rate $HR$, false alarm ratio $FAR$ and critical
success index $CSI$ as defined in (Bennett et al., 2013). In addition, we evaluated the total area flooded above a water level of
0.1m.





## 3.3  C - Flood damage assessment

Based on the 2D flood simulations performed for the baseline situation, we assessed flood damages. The derived damage data were subsequently used as a reference for training and validating the regression models derived in Sect. 3.4.

Direct flood damages in urban areas are commonly assessed by overlaying polygons of exposed objects with high-resolution

flood maps. A damage is then assigned to each object (e.g., a building) depending on the greatest adjacent water depth (Hammond et al., 2015). For our assessment, we have focused on direct, tangible flood damages as these are most directly related to the urban form.

We distinguished two approaches for damage assessment, which we expected might yield different results in terms of which impacts different data resolutions may have in damage assessment. The first type are threshold-based approaches, where a unit-

190 damage is assigned to an object if the water level exceeds a defined threshold. In Denmark, such approaches are frequently applied in the context of pluvial risk assessments (Kaspersen and Halsnæs, 2017; Odense Kommune, 2014; Olsen et al., 2015), because water levels are generally low. In the international literature, depth-damage curves are widely applied (Penning-Rowsell et al., 2013; Thieken et al., 2008), where damage potentials are assigned to different objects in the urban space. Depending on the flood water level, different portions of the damage potential are realized.

We considered the framework of Olsen et al. (2015) as an example for the unit-damage approach, while the framework of Beckers et al. (2013) was considered as an example for the depth-damage based approach. The latter builds on damage functions from FLEMO. It is the only example we were able to find in the literature where damage potentials for residential and commercial properties were published for the same case study. We have therefore selected it for our work. Table 1 summarizes both approaches. We have not considered damages to road structures, because these were of negligible magnitude.

Flood damages were derived by overlaying the simulated flood areas with the building polygons. A damage per sqm was derived for each building, considering the damage functions shown in Table 1. The building polygons were then rasterized to a resolution of 1m and subsequently aggregated to the the different data resolutions used for fitting the regression models detailed in Sect. 3.4.

We have also derived flood damages for the baseline simulation where buildings were not included in the DEM. The damage

values were not used for regression, but are shown in the results section, as they provide insight on the impact of blocked flow paths on damage assessment.

## 3.4  D - Flood damage regression

### 3.4.1  Model setup

In the regression of flood damages, we considered the building footprint area $A_{flooded,WL[i]}$ flooded with a water level above

210 threshold $WL[i]$ as the main input variable. This area was determined by down-sampling the building raster data with resolutions of 25, 50, 100, 200, 300, 500, 750 and 1000m to the same resolution as the flood maps (5m) and summing up the building areas for all pixels which were flooded above the threshold of interest.





We reasoned that the regression models should reflect the characteristics of the damage function applied in the original damage assessment. We have therefore considered a model structure based on the three building classes considered in damage

assessment. A square-root transformation was applied to both input and output variables based on an analysis of scatter plots between inputs and outputs:

$$D^{0.5} = \sum_{i=1}^{n}(b_{1i}A_{flooded,res,WL[i]}^{0.5}+$$
$$b_{2i}A_{flooded,comm,WL[i]}^{0.5}+$$
$$b_{3i}A_{flooded,pub,WL[i]}^{0.5}). \quad (6)$$

The flooded building footprint areas for residential ($A_{flooded,res}$), commercial ($A_{flooded,pub}$) and public ($A_{flooded,pub}$) buildings were computed as the total footprint area of the corresponding class that was flooded above water level $WL[i]$

and below $WL[i+1]$. The mapping between the 11 building types considered in our case study and three building classes considered for damage assessment is illustrated in Table S1.

Both, for the damage data derived based on Olsen et al. (2015) and on Beckers et al. (2013), we have applied Eq. (6) with a single damage threshold of 0.1m, resulting in a model with three input variables that corresponded to the total flooded footprint area for each building type. This approach was in the following named **DMOD1**. In addition, for the damage data derived

based on Beckers et al. (2013), we also applied a model where all five water level thresholds shown in Table 1 were considered. The result was a regression model with 15 input variables that reflected the building footprint areas flooded above the different water level thresholds considered in the original damage assessment. This approach was called **DMOD2**.

Similar to the approach for impervious areas in Sect. 3.1, we fitted the regression models DMOD1 and DMOD2 considering 80 different input data resolutions between 25 and 2000m. The flooded building area $A_{bf,WL[i]}$ was always determined at a

230 resolution of 5m (corresponding to the resolution of the flood map), and was subsequently aggregated to the resolution that should be used for regression fitting.

To distinguish to what extent coarse building data affect damage assessment by creating uncertainty on flood exposure or flood hazard, we derived flooded building areas both from the baseline flood map and from the flood map created in a 2D simulation with the aggregated building data which were also considered for damage regression.

**3.4.2 Performance assessment**

To assess model performance, we performed cross validation. The city was divided into subareas of 2000x2000m (see Sect. S5 in the supporting information). We trained the regression model on a random sample of 80% of the subareas and assessed model performance on the remaining 20%. This process was repeated $k = 1000$ times.

When the regression models were fitted to datasets with resolutions finer than 2000m, we linked the pixels at the lower data

resolution to the subarea with which they overlapped most. Regression modeling was then performed at the finer resolution, and predicted damages for each subarea were computed by aggregating the values from the linked pixels. The subdivision into subareas allowed us to evaluate model performance at a constant spatial scale despite applying different data resolutions for



model fitting. However, it had the disadvantage that the pixels in the datasets used for regression modeling were not always completely included in a subarea, leading to noise in the computed scores.

To evaluate regression fit, we computed the NSE of damage values $D_{pred,j,k}$ predicted for each subarea $j$ in the validation dataset by comparing against the baseline damage $D_{baseline,j}$ value for the same subarea:

$$NSE_{D,CV2000,k} = 1 - \frac{\sum_j \left(D_{pred,j,k} - D_{baseline,j}\right)^2}{\sum_j \left(D_{baseline,j} - \overline{D_{baseline}}\right)^2} \tag{7}$$

In addition, we computed the total damage ratio $DR_{tot,k}$ considering all subareas $j$ in the validation dataset as

$$DR_{tot,k} = \sum_j D_{pred,j,k} / \sum_j D_{baseline,j}. \tag{8}$$

Median values of $NSE_{D,CV2000,k}$ and $DR_{tot,k}$ were considered in the analysis of results. For the cases where flooded building areas $A_{flooded,WL[i]}$ were derived based on the flood map from the baseline simulation, scores were marked with subscript **BF**.

## 4    Results

The results section was structured into the same parts that were also highlighted in Fig. (3). Performance scores related to the
simulation of flood hazards and the assessment of flood damages (parts B to D) were collected in Tables 2 and 3, distinguishing results for building data with varying resolutions.

### 4.1    A - Estimation of impervious areas

Figure 4 summarizes $NSE_{Aimp,k}$, $RBIAS_{Aimp,k}$ and $RMSE_{Aimp,k}$ where regression models for impervious area were fitted for varying data resolutions ($\Delta x_{fit}$), and where the coefficients fitted for one resolution were used to predict impervious
areas considering building data aggregated to varying resolutions as input ($\Delta x_{pred}$). Subfigures A-C show histograms of the score values obtained during 1000 cross validation iterations for the combination $\Delta x_{fit} = \Delta x_{pred} = 500m$, while subfigures D-F show median values of $NSE_{Aimp,k}$ and $RBIAS_{Aimp,k}$ and the standard deviation of $RMSE_{Aimp,k}$ obtained for each of the $80 \cdot 80$ combinations of $\Delta x_{fit}$ and $\Delta x_{pred}$.

When the regression models were fitted to data with resolutions below approximately 250m, the relationship between build-
ing footprint areas and imperviousness could not be identified, because building footprint areas would then not necessarily be located in the same pixels as the associated features of the urban layout (e.g., sidewalks). This lead to low values for $NSE_{Aimp}$ and an under-prediction of the total imperviousness ($RBIAS_{Aimp} < 1$). Values of $NSE_{Aimp}$ above 0.95 were achieved when predicting impervious areas at spatial scales above 500m. For finer spatial scales, there would be random variations in the imperviousness that could not be explained by the amount of building footprint areas alone (see also Fig. (S1)).
While the median predictive performance of the regression models ($NSE_{Aimp}$ and $RBIAS_{Aimp}$) remained constant for fitting resolutions between approximately 250 and 2000m, the standard deviation of the $RMSE$ values obtained for a fixed


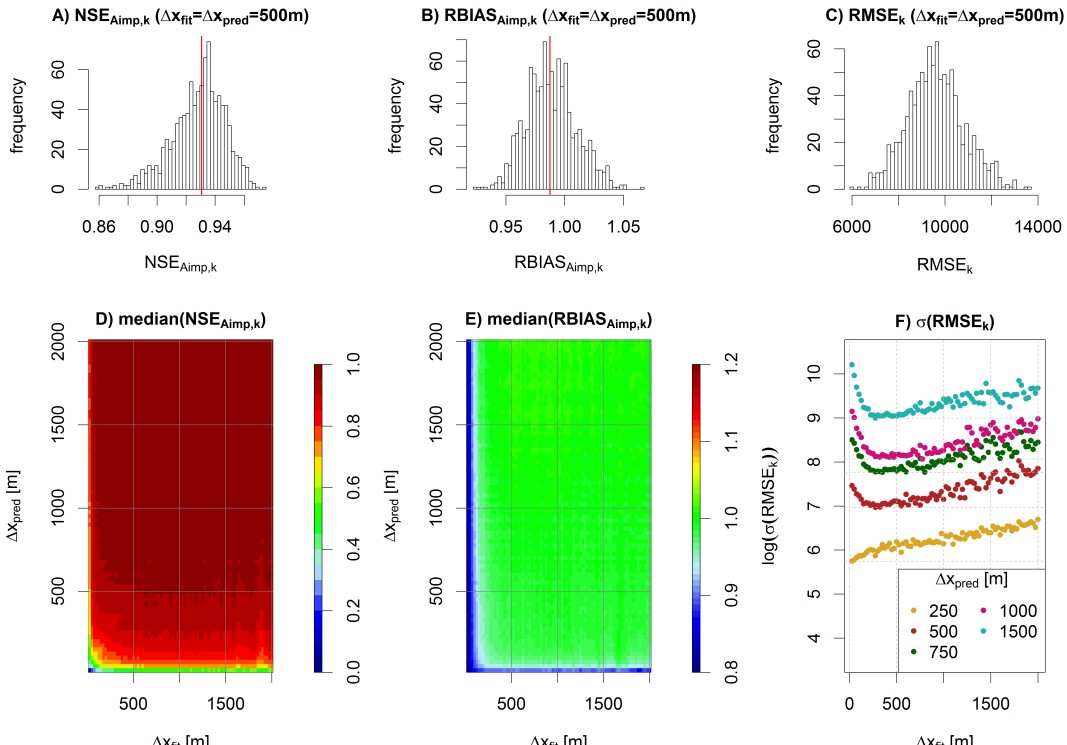

**Figure 4.** Subfigures A-C: Histograms of $NSE_{Aimp,k}$, $RBIAS_{Aimp,k}$ and $RMSE_k$ obtained during 1000 cross validation iterations $k$ for the combination of fitting resolution $\Delta x_{fit} = 500m$ and prediction resolution $\Delta x_{pred} = 500m$. Red lines in subfigures A and B indicate median values. Subfigures D and E: median values of $NSE_{Aimp,k}$ and $RBIAS_{Aimp,k}$ obtained for varying combinations of $\Delta x_{fit}$ and $\Delta x_{pred}$. Subfigures F: standard deviation (log-transformed) of $RMSE_k$ obtained for varying values of $\Delta x_{fit}$ and selected values of $\Delta x_{pred}$ (dots with varying colours).

prediction resolution was minimal for fitting resolutions in the order of 400m, i.e., for coarser fitting resolutions there would be a larger portion of the cross validation iterations where the regression models would not be properly identified. This behavior was considered plausible, because coarser fitting resolution are accompanied by a loss of information on spatial variability, and because the decreasing number of data points may make it harder to identify the models. Thus, for our case study, we identified a fitting resolution of 400m as the optimal trade-off between capturing the link between urban layout and imperviousness by data aggregating data into large enough pixels, and avoiding loss of information by blurring the dataset.

## 4.2 B - 2D flood simulation

Figure 5 shows the total area which was simulated flooded above different water level thresholds. Results are compared for the baseline model and for a model where imperviousness was specified based on building footprint areas aggregated to a raster resolution of 200m. The figure suggests that the model based on aggregated building data simulated fewer areas flooded with

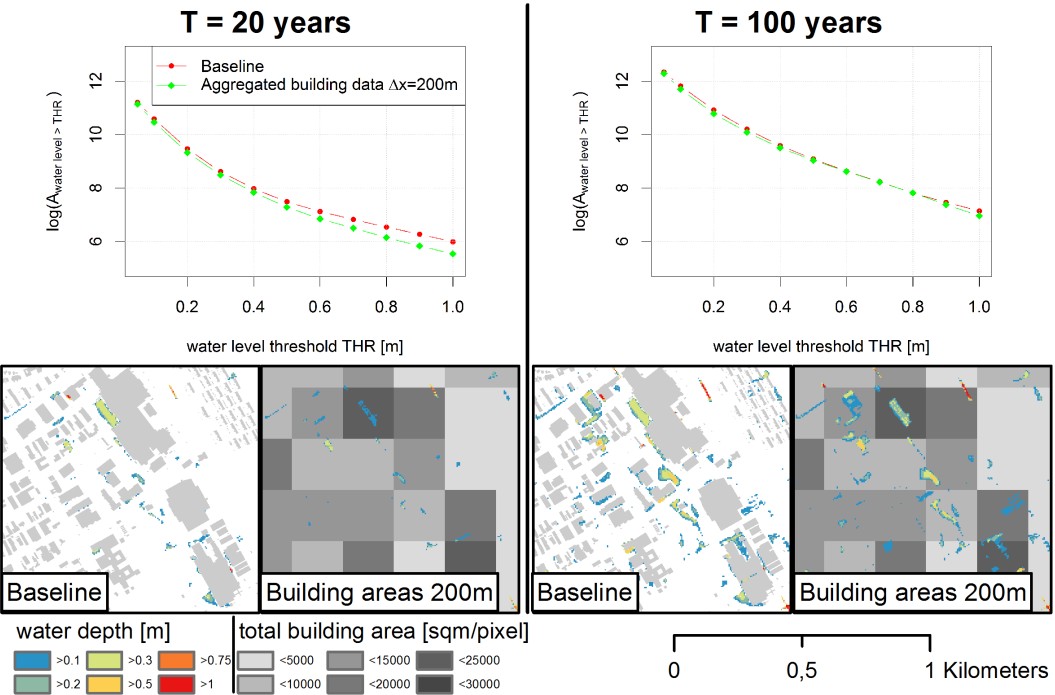

**Figure 5.** Total area flooded above water level threshold in baseline 2D simulation and in simulation based on building footprint areas aggregated to 200m raster. Results are shown for return periods of 20 (left) and 100 (right) years. Maps below the plots illustrate simulated water depths in the different cases with background showing building footprint polygons (baseline) and total building footprint area per 200x200m pixel (aggregate building data).

high water levels for the 20 year event. The reason was that this model did not consider the blockage of surface flow paths by buildings. The effect can also be seen by comparing the flood maps in the lower part of Fig. (5).

For the 100 year event, similar total flooded areas were obtained for both models, which can be associated to the greater degree of water movement on the surface and, as a result, the filling of sinks in both models. However, the performance scores shown in Table 3 suggest that there was substantial disagreement between the two models in where flooding occurred. It was difficult to conclude how severely simulated flood maps deviated from the baseline in absolute terms because the performance scores were based on pixel by pixel comparisons, and thus suffered from double penalty issues.

For both return periods, the score values in Tables 2 and 3 suggest that the flood maps generated with models based on aggregated building data generally resembled the flood map from the baseline simulation without buildings. An increasingly coarse representation of imperviousness in the model thus had little impact on the simulated flood maps as compared to the effect caused by the blockage of flow paths in the baseline simulation.

A minor effect was noticeable in particular in the total simulated flood areas. Coarse building area resolutions implied that impervious areas would be distributed increasingly evenly over the catchment, leading to the distribution of effective


precipitation over larger areas , surface flows with small water levels and, as a result, fewer areas where water levels would exceed the threshold of 0.1m. On the other hand, total impervious areas would be underestimated by the regression model for fine building datasets as a result of the regression specification without intercept. In fact, total impervious areas would be underestimated by 10% with the 25m building raster set, while the bias would exponentially decrease to under 1% at a resolution of 300m. These two competing effects implied that the flood maps obtained based on 25m building raster data

resembled the baseline best in the 20 year event, where runoff depths were comparably small, and significant water depths only occurred due to an aggregation of impervious areas. For the 100 year event, raster sets with resolutions of 50 and 100m yielded the best trade-off between avoiding an underestimation of impervious areas and ensuring sufficient spatial aggregation of impervious areas.

It needs to be emphasized that the effects discussed above were very minor compared to the impact of whether buildings were

considered in the DEM applied for 2D simulation or not. The missing impact of increasingly coarse representations of imperviousness is likely to be linked to the fact that sewer systems were considered by reducing effective rainfall in a manner which was proportional to the imperviousness in a pixel (Eq. (5)), i.e., the design of the assumed sewer system followed the distribution of impervious areas in space.

### 4.3   C - Damage assessment

Figure 6 compares damages derived based on the baseline flood map and based on the flood map where buildings were not considered in the 2D simulation of surface flows. In general, the latter approach lead to an underestimation of flood damages, because blocked flow paths in the baseline lead to higher water levels.

The figure also illustrates differences in the results obtained for the two damage frameworks. Considering an aggregation level of 400m, we noticed individual pixels where damages derived using depth-damage curves (Beckers et al., 2013) were

several times greater than for the threshold-based method (Olsen et al., 2015), while damages were of similar magnitude on an aggregation level of 2000m. In addition, the approach based on depth-damage curves was subject to stronger variations and and stronger underestimation of total damages. These effects were mainly caused by large commercial buildings which could induce very high damage values when water ponded next to these buildings in the baseline simulation, even though the flooded area would often be small. The threshold-based damage assessment was more robust towards such effects, because a

unit damage would be assigned which depended on neither the building size nor the water level.

### 4.4   D - Damage regression

Performance scores for damage regression models fitted based on building data with varying aggregation levels were summarized in Tables 2 and 3. The scores shown in the tables were derived considering a data resolution $\Delta x_{fit}$ of 1000m.

The damage regression generally scored high values for $NSE_{D,CV2000}$ (median values obtained in cross validation) and

only slightly biased total damages ($DR_{tot}$), suggesting that, on aggregation levels of 2000m and above, the regression models were able to compensate for deviations in both the simulated flood area and for aggregated representations of building exposure in the form of raster representations of building footprint areas. In addition, there was little difference in the regression scores


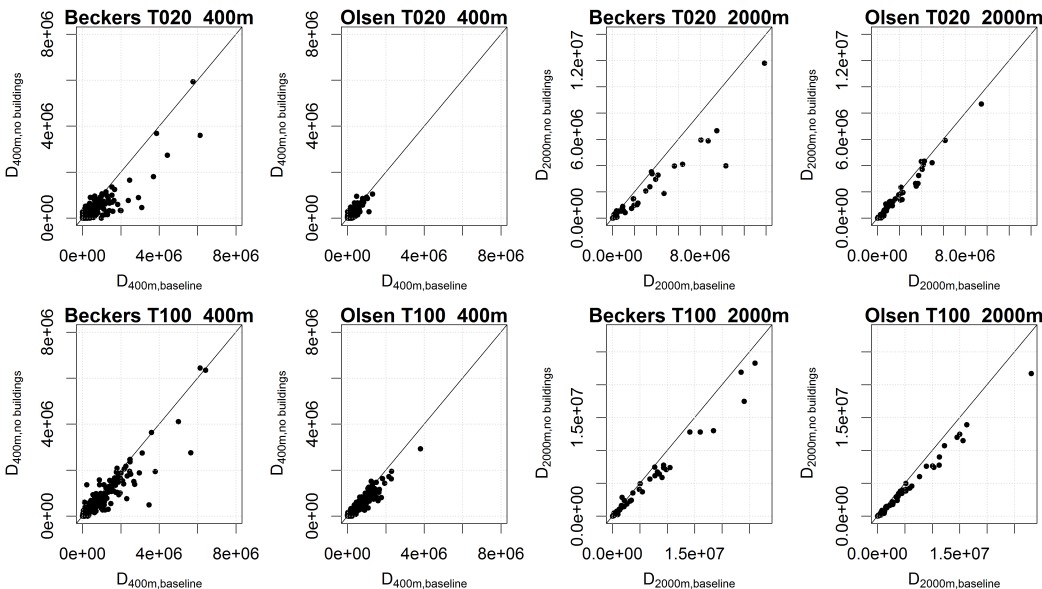

**Figure 6.** Scatterplots of flood damages estimated based on 2D flood simulations with (baseline) and without buildings included in DEM. Results are shown for return periods of 20 (top row) and 100 (bottom row) years, for both damage assessment frameworks and for spatial aggregation levels of 400 and 2000m. Damages were assessed by overlaying building polygons and the corresponding flood areas.

when flooded building areas were derived using flood maps created based on the aggregated building data, and when the baseline flood map was applied (comparing $NSE_{D,CV2000}$ and $NSE_{D,CV2000,BF}$), supporting the statement above.

Both of the above statements were not true for the cases where damages were derived based on the framework of Beckers et al. (2013) in the 20 year event. Similar to the observations in Sect. 4.2 and 4.3, this effect was tied to local ponding near large buildings in the baseline simulation and the associated large damages assigned by the framework of Beckers et al. (2013). $NSE_{D,CV2000,BF}$ was much higher in these cases than $NSE_{D,CV2000}$ which underlines that the regression models were not able to reproduce damages simply because no or insufficient degrees of flooding were simulated in areas where major damages

occurred.

Figure 7 illustrates how $NSE_{D,CV2000}$ varied when different data resolutions were applied for regression model fitting ($\Delta x_{fit}$), and when different building data resolutions were considered for both parametrizing imperviousness in the surface flood simulations and for computing flooded building area as input to the regression models. As the computed score values were noisy (see Sect. 3.4), we have displayed smoothed lines (R function "'loess"' with parameter $span = 0.25$ (Cleveland

et al., 1992; R Core Team, 2018)). True values were included as dots to illustrate the level of variation around the smoothed line. Values obtained for the best performing building data resolution of 200m were colored blue.

Similar to the the results obtained for impervious areas, a minimal data resolution between 200 and 1000m was required to properly identify the regression models, depending on the damage framework and the resolution of the building data considered. More surprisingly, building data with a resolution of 200m consistently yielded high $NSE_{D,CV2000}$ values, while


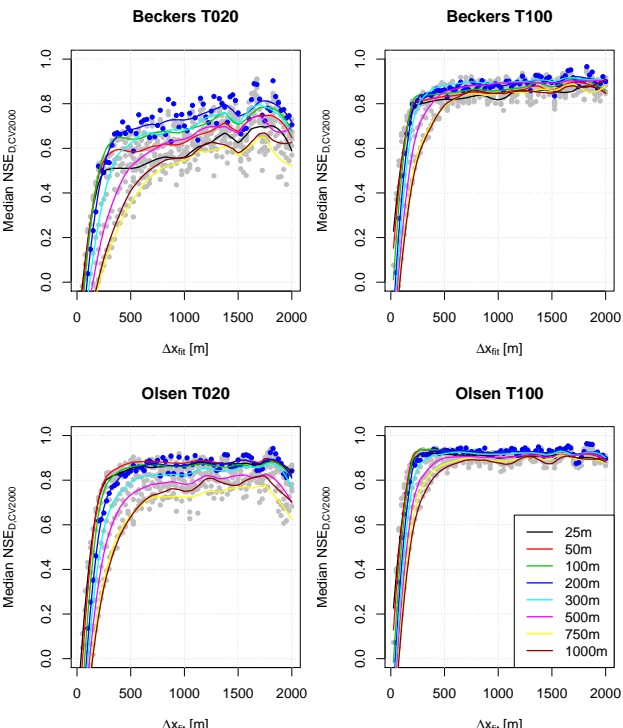

**Figure 7.** $NSE_{D,CV2000}$ considering flood damage regression models (DMOD1) fitted at different data resolutions ($\Delta x_{fit}$) and considering building data aggregated to different resolutions in m (lines with varying colors). Lines were smoothed while dots indicate the true $NSE_{D,CV2000}$ values derived for each combination of fitting resolution and building input data resolution. Dots were colored blue for a building data resolution of 200m and grey otherwise.

high resolution building data only yielded high score values when damages were computed according to the threshold-based approach of Olsen et al. (2015).

Figure 8 illustrates for the framework of Beckers et al. (2013) and a return period of 100 years the damages computed in the baseline simulation, and compares them against regression predictions generated using building data aggregated to raster resolutions of 25, 200 and 750m. For a building data resolution of 25m substantial over- and under-predictions of damages were observed. These effects were mediated when considering coarser building data with a resolution of 200m, while the coarsest building dataset with a resolution of 750m no longer allowed to capture the spatial variability of flood damages.

Figure 2 illustrates simulated flood areas and building data for the pixels marked as "'Area 1"' and "'Area 2"' in Fig. (8). Similar damages were observed in the baseline simulation for both areas. However, the extent of the flooded area is very different in both cases. In particular, only very small parts of the building overlap with the flooded area in area 2 for a building data resolution of 25m. For a data resolution of 200m, the spatial averaging of building areas leads to a lower value for the flooded building area in area 1, and a higher value in area 2, allowing for a better regression fit. Similar to the discussion in

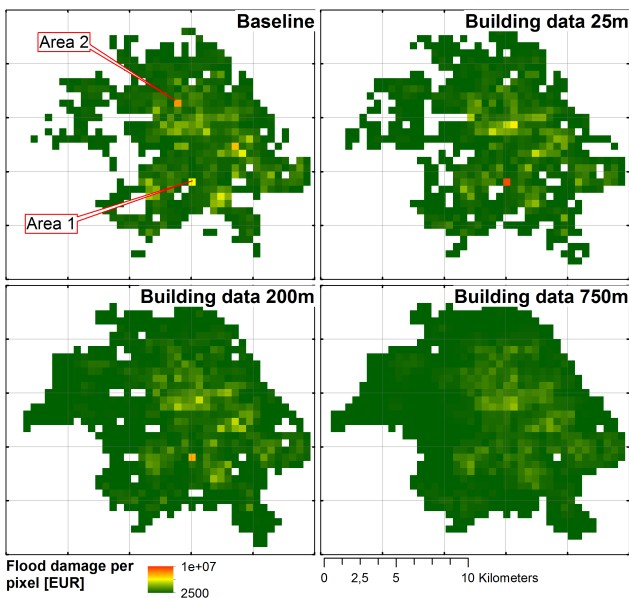

**Figure 8.** Flood damages predicted by DMOD1 on an aggregation level of 500, considering the baseline dataset and regression predictions generated with building data aggregated to resolutions of 25, 200 and 750m. Damages were computed using the framework documented by Beckers et al. (2013) for a return period of 100 years. Flood areas and building data for the pixels named area 1 and 2 are shown in Fig. (2).

Sect. 4.3, this effect was less pronounced when flood damages were computed according to the framework of Olsen et al. (2015), because the threshold-method was less prone to creating high damages in individual locations.

Finally, comparing values of $NSE_{D,CV2000}$ and $DR_{tot}$ for DMOD1 and DMOD2 in Tables 2 and 3, little difference could be observed between the two models. In fact, the more complex DMOD2 occasionally yielded lower scores, because more parameters needed to be identified. In addition, the flooded building areas for different level thresholds were correlated, because areas with high water depths would typically also be associated with greater flood extents in general (Fig. (2)), and the additional variables thus yielded little additional information in the regression process.

# 5 Discussion

## 5.1 Using aggregated building data for flood risk assessment

The results suggest that the consideration of aggregated building data affected both the simulation of flood hazards, and the assessment of flood damages. In terms of the simulated flood hazards, the main effect arose from not considering the blockage of surface flow paths in the 2D flood simulations when considering aggregated building data, while coarse representations of imperviousness had little effect.





Despite the aggregation of building data, we were able to achieve realistic representations of flood exposure, which was illustrated by the high $NSE_{D,CV2000}$ and $DR_{tot}$ values obtained during damage regression. Building data aggregated to resolutions in the order of 200m yielded better regression performance than building data with finer resolutions when considering damages derived using the depth-damage based framework of Beckers et al. (2013). Performance of the finer and coarser datasets was similar when considering damages derived based on the threshold-based framework of Olsen et al. (2015). These

trends were independent on whether the baseline flood map was applied in damage regression, or the flood map simulated based on aggregated building data. Slightly higher aggregation levels of the building raster sets can thus be considered beneficial for flood screening approaches, as it yields a more robust representation of flood exposure.

    The damage regression yielded total damage estimates that, for a building data resolution of 200m, differed between 1 and 10% from the baseline values. This was considerably better than the total damage values obtained in the baseline simulation

where buildings were neglected in the 2D flood simulation, but damage assessment was performed using building polygon data. This highlights the need for adjusting damage frameworks developed for high-resolution data to the actual modeling context.

### 5.2   Damage frameworks for pluvial flood risk assessment

The damage assessment approach based on depth-damage curves (Beckers et al., 2013) produced high, localized damage values where flow paths were blocked by large buildings. These situations were difficult to reproduce using aggregated building data,

because it was not possible to simulate the local ponding of water, in particular for the smaller event.

    It is questionable whether this damage assessment approach is reasonable for pluvial flood risk, because it relies on modeled water depths which in reality would be unlikely to occur in this form, because the water would likely enter the building and distribute without causing major structural damages. Damage assessment approaches which are less sensitive to water depths may thus be preferable for pluvial flood risk assessment.

The issue could be mitigated by explicitly considering water flow through buildings in the surface flow model, which, however, poses technical challenges. Alternatively, robust regression approaches are likely to yield better results when performing damage regression in the presence of such issues.

### 5.3   Data resolution in the development of scaling approaches

Very clear dependencies on spatial scale could be identified when developing regression models that predicted impervious

areas as a function of building footprint areas. The optimal data resolution for developing these models was identified to be in the order of 400m. For finer data resolutions, buildings would not necessarily be located in the same pixel as other impervious areas linked to the buildings (e.g., sidewalks), resulting in an underestimation of impervious areas by the regression models. For coarser resolutions, the data would gradually become too aggregated to properly identify the link between the different building types and imperviousness, leading to a stronger variability of the results during cross-validation. Reliable predictions

of imperviousness could be obtained at spatial scales above 500m (NSE>0.95).

    In a similar manner, the performance of regression models for flood damages only reached acceptable levels when data resolutions between 500 and 1000m were considered during parameter estimation, depending on the level of aggregation of



the considered building dataset. $DR_{tot}$ approached values near 1 only when data resolutions $\Delta x_{fit}$ of 1500m and coarser were considered (see Figure S4), suggesting that the data needed to be aggregated to such levels to counterbalance local variations in where flooding was simulated and which buildings were exposed to flooding.

## 5.4 Limitations

We performed 2D surface flow simulations based on publicly available DEM data where buildings and plants were removed in an automated manner. Our results suggest that the simulated flood maps were very strongly affected by whether the blockage of flow paths through buildings was considered in the DEM or not. Remnants originating from the DEM cleaning process may affect this result and could be an explanation for the rather low performance scores of the simulations where buildings were not included in the DEM. For example, slight misalignments between building polygons and building locations in the DEM may result in artificial sinks in the baseline simulation which would not be possible to reproduce in simulations without buildings.

Our 2D flood modeling approach was a simplified representation of the urban water cycle. This approach was justified as our intention was to evaluate which spatial scales should be considered in the development of flood screening approaches. For detailed assessment of the risk we would recommend 1D-2D calculation methods to more accurately represent where flooding occurs in the catchment.

Finally, the regression models for imperviousness and flood damages are likely to depend on topography and urban layout. Thus, different models would need to be trained for different case studies, while we expect that the scale dependencies identified in our work are generic. More importantly, if landuse planning is implemented in a flood-aware manner, the relationship between flood damages and the flooded building area computed from aggregated data will change. This effect can be considered by training regression models to different datasets, which is an important line of future research in the development of flood screening approaches.

## 6   Conclusions

We studied how different data resolutions affect the identification of empirical relationships between building data and urban hydrology, and at which spatial scales reasonable predictions could be obtained. Based on our results, we draw the following conclusions:

1. The identification of empirical relations between urban layout and urban hydrology is subject to a bias-variance-tradeoff. Too fine spatial data resolutions prevent the identification of empirical relationships and lead to biased results, while too coarse resolutions reduce the number of data points and blur out spatial variations, leading to uncertainty in the estimated relationships.

2. Simulated pluvial flood hazards are strongly affected by whether surface flow simulations consider the blockage of flow paths through buildings, and less by spatially averaged representations of imperviousness during rainfall runoff calculations.



3. Water levels are underestimated if local ponding near buildings is not considered in the surface flow simulations, as would be the case when considering aggregated building data for flood screening. Without correction, this effect also leads to an underestimation of flood damages.

4. A simple regression model predicting flood damages in an area as a function of the amount of flooded building area can, to some extent, compensate for deficiencies in the simulated flood area. Building data aggregated to resolutions in the order of 200m are the preferred input and perform more robust than building data with finer resolutions, because they reduce local extrema in flooded building areas.

5. Regression models for flood damage must be expected to depend on whether flood-aware spatial planning was applied in the case study used for model training or not. Different models must thus be trained to consider different land-use management strategies.

6. Local ponding next to large buildings can create rather large water levels in simulations of pluvial flood risk that may be unrealistic. Damage assessment frameworks where damages increase as a function of water levels are vulnerable to this type of error which is specific to pluvial flood risk.

*Code and data availability.* Computer code for fitting regression models for imperviousness and flood damages was made available by Löwe (2019). Building and imperviousness data were proprietary. These datasets and the derived 2D flood models were therefore not made available. Upon request, the authors will attempt to obtain permission for sharing this data.

*Author contributions.* Roland Löwe performed the analysis and led the preparation of the manuscript. Karsten Arnbjerg-Nielsen supported the scoping of the study and provided feedback on various iterations of the results and the manuscript.

*Competing interests.* We declare no conflict of interest.

*Acknowledgements.* This project was funded by Innovation Fund Denmark through the Water Smart Cities Project (Grant no. 5157-00009B). We wish to thank Odense municipality and Vandcenter Syd (VCS Denmark) for the provision of data used in this study. In particular, we wish to thank Agnethe N. Pedersen and Nena Kroghsbo for their support, feedback and discussions.



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





**Table 1.** Damage assessment frameworks considered in our work

| type | Olsen et al. (2015) | | Beckers et al. (2013) | | | | |
|---|---|---|---|---|---|---|---|
| | | | | immobile | | mobile | |
| | WL [m] | unit damage [EUR] | WL [m] | loss ratio [%] | damage potential [EUR/ sqm] | loss ratio [%] | damage potential [EUR/ sqm] |
| residential | >0.1 | 1800 | >0.1 | 3 | | 3 | |
| | | | >0.21 | 8 | | 8 | |
| | | | >0.6 | 11 | 389 | 11 | 119 |
| | | | >1.0 | 17 | | 17 | |
| | | | >1.5 | 22 | | 22 | |
| public | >0.1 | 8300 | >0.1 | 5 | | 29 | |
| | | | >0.21 | 9 | | 30 | |
| | | | >0.6 | 17 | 370 | 42 | 1.32 |
| | | | >1.0 | 23 | | 48 | |
| | | | >1.5 | 39 | | 61 | |
| commercial | >0.1 | 9500 | >0.1 | 5 | | 29 | |
| | | | >0.21 | 9 | | 30 | |
| | | | >0.6 | 17 | 343 | 42 | 90 |
| | | | >1.0 | 23 | | 48 | |
| | | | >1.5 | 39 | | 61 | |



**Table 2.** Summary scores for return period T=20 years. The top section compares flood areas simulated with the varying input datasets against the baseline simulation. The middle and lower section evaluate goodness of fit for the damage regression models, considering separate results for the two damage assessment frameworks. Values shown for $NSE_D$ and $DR_{tot}$ correspond to median values obtained during cross validation. The subscript BF marks those cases were flood areas from the baseline simulation were used to determine flooded building areas for regression. Score values for damage regression were derived at a fitting resolution $\Delta x_{fit} = 1000m$.

| | Score | Baseline, buildings excluded from DEM | Aggregated building footprint areas used for predicting imperviousness and for damage regression - resolution in m | | | | | | | |
| --- | --- | --- | --- | --- | --- | --- | --- | --- | --- | --- |
| | | | 25 | 50 | 100 | 200 | 300 | 500 | 750 | 1000 |
| Comparison of | $CSI$ | 0.55 | 0.55 | 0.55 | 0.55 | 0.55 | 0.55 | 0.55 | 0.55 | 0.54 |
| simulated flood | $HR$ | 0.70 | 0.69 | 0.68 | 0.67 | 0.67 | 0.67 | 0.66 | 0.67 | 0.67 |
| areas | $FAR$ | 0.27 | 0.26 | 0.25 | 0.24 | 0.24 | 0.24 | 0.25 | 0.25 | 0.26 |
| (WL>0.1m) | $A_{flood}/A_{flood,baseline}$ | 0.96 | 0.92 | 0.90 | 0.89 | 0.88 | 0.88 | 0.88 | 0.89 | 0.90 |
| | DMOD1-$NSE_{D,CV2000}$ | | 0.54 | 0.58 | 0.66 | 0.7 | 0.64 | 0.67 | 0.59 | 0.64 |
| Flood damage | DMOD1-$NSE_{D,CV2000,BF}$ | NSE=0.84[1] | 0.74 | 0.86 | 0.89 | 0.88 | 0.86 | 0.79 | 0.72 | 0.73 |
| assessment | DMOD2-$NSE_{D,CV2000}$ | | 0.52 | 0.63 | 0.66 | 0.65 | 0.47 | 0.7 | 0.67 | 0.74 |
| Beckers et al. | DMOD2-$NSE_{D,CV2000,BF}$ | | 0.77 | 0.87 | 0.88 | 0.86 | 0.75 | 0.75 | 0.7 | 0.72 |
| (2013) | DMOD1-$DR_{tot}$ | | 0.85 | 0.86 | 0.88 | 0.9 | 0.85 | 0.85 | 0.74 | 0.84 |
| | DMOD1-$DR_{tot,BF}$ | DR=0.68[1] | 0.93 | 0.92 | 0.94 | 0.95 | 0.91 | 0.91 | 0.8 | 0.87 |
| | DMOD2-$DR_{tot}$ | | 0.87 | 0.88 | 0.91 | 0.93 | 1.01 | 0.96 | 0.96 | 0.97 |
| | DMOD2-$DR_{tot,BF}$ | | 0.91 | 0.93 | 0.96 | 0.93 | 0.95 | 0.97 | 1.03 | 1.02 |
| Flood damage | DMOD1-$NSE_{D,CV2000}$ | NSE=0.96[1] | 0.88 | 0.89 | 0.87 | 0.86 | 0.82 | 0.79 | 0.75 | 0.76 |
| assessment | DMOD1-$NSE_{D,CV2000,BF}$ | | 0.88 | 0.88 | 0.86 | 0.84 | 0.82 | 0.82 | 0.77 | 0.8 |
| Olsen et al. | DMOD1-$DR_{tot}$ | DR=0.88[1] | 0.92 | 0.96 | 0.96 | 0.95 | 0.94 | 0.94 | 0.9 | 0.91 |
| (2015) | DMOD1-$DR_{tot,BF}$ | | 0.95 | 0.96 | 0.97 | 0.96 | 0.96 | 0.95 | 0.93 | 0.93 |

[1] $NSE$ was computed by aggregating damages derived for the baseline simulation without buildings to 2000m, and comparing these results against the baseline simulation with buildings. $DR$ was computed by computing the ratio of total damages in both simulations.



**Table 3.** Summary scores for return period T=100 years. The top section compares flood areas simulated with the varying input datasets against the baseline simulation. The middle and lower section evaluate goodness of fit for the damage regression models, considering separate results for the two damage assessment frameworks. Values shown for $NSE_D$ and $DR_{tot}$ correspond to median values obtained during cross validation. The subscript BF marks those cases were flood areas from the baseline simulation were used to determine flooded building areas for regression. Score values for damage regression were derived at a fitting resolution $\Delta x_{fit} = 1000m$.

| | Score | Baseline, buildings excluded from DEM | Aggregated building footprint areas used for predicting imperviousness and for damage regression - resolution in m | | | | | | | |
|---|---|---|---|---|---|---|---|---|---|---|
| | | | 25 | 50 | 100 | 200 | 300 | 500 | 750 | 1000 |
| Comparison of simulated flood areas (WL>0.1m) | $CSI$ | 0.59 | 0.60 | 0.59 | 0.59 | 0.59 | 0.59 | 0.59 | 0.59 | 0.59 |
| | $HR$ | 0.73 | 0.71 | 0.71 | 0.71 | 0.70 | 0.70 | 0.70 | 0.70 | 0.70 |
| | $FAR$ | 0.24 | 0.21 | 0.23 | 0.23 | 0.21 | 0.21 | 0.20 | 0.20 | 0.20 |
| | $A_{flood}/A_{flood,baseline}$ | 0.95 | 0.90 | 0.93 | 0.92 | 0.89 | 0.88 | 0.88 | 0.88 | 0.88 |
| Flood damage assessment Beckers et al. (2013) | DMOD1-$NSE_{D,CV2000}$ | NSE=0.94[1] | 0.86 | 0.86 | 0.86 | 0.88 | 0.91 | 0.91 | 0.86 | 0.91 |
| | DMOD1-$NSE_{D,CV2000,BF}$ | | 0.89 | 0.91 | 0.91 | 0.91 | 0.93 | 0.91 | 0.88 | 0.9 |
| | DMOD2-$NSE_{D,CV2000}$ | | 0.78 | 0.76 | 0.83 | 0.88 | 0.89 | 0.88 | 0.91 | 0.86 |
| | DMOD2-$NSE_{D,CV2000,BF}$ | | 0.87 | 0.89 | 0.89 | 0.9 | 0.89 | 0.86 | 0.88 | 0.85 |
| | DMOD1-$DR_{tot}$ | DR=0.81[1] | 0.92 | 0.94 | 0.93 | 0.95 | 0.93 | 0.95 | 0.9 | 0.94 |
| | DMOD1-$DR_{tot,BF}$ | | 0.96 | 0.97 | 0.95 | 0.96 | 0.95 | 0.94 | 0.92 | 0.95 |
| | DMOD2-$DR_{tot}$ | | 0.93 | 0.97 | 0.95 | 0.94 | 0.94 | 0.97 | 0.95 | 0.99 |
| | DMOD2-$DR_{tot,BF}$ | | 0.95 | 0.98 | 0.96 | 0.97 | 0.96 | 0.96 | 0.95 | 0.99 |
| Flood damage assessment Olsen et al. (2015) | DMOD1-$NSE_{D,CV2000}$ | NSE=0.94[1] | 0.9 | 0.91 | 0.92 | 0.92 | 0.92 | 0.9 | 0.88 | 0.9 |
| | DMOD1-$NSE_{D,CV2000,BF}$ | | 0.94 | 0.93 | 0.93 | 0.93 | 0.91 | 0.92 | 0.89 | 0.91 |
| | DMOD1-$DR_{tot}$ | DR=0.79[1] | 0.95 | 0.97 | 0.99 | 0.98 | 0.98 | 0.97 | 0.96 | 0.97 |
| | DMOD1-$DR_{tot,BF}$ | | 0.98 | 0.97 | 0.99 | 0.98 | 0.97 | 0.98 | 0.97 | 0.98 |

[1] $NSE$ was computed by aggregating damages derived for the baseline simulation without buildings to 2000m, and comparing these results against the baseline simulation with buildings. $DR$ was computed by computing the ratio of total damages in both simulations.