# Peer review of "Urban pluvial flood risk assessment - data resolution and spatial scale when developing screening approaches on the micro scale"

_Natural Hazards and Earth System Sciences, 2019_

## Referee Comment (RC1) · Anonymous Referee #1 · 25 Sep 2019

**Review of manuscript „Urban pluvial flood risk assessment - data resolution and spatial scale when developing screening approaches on the micro scale" by Roland Löwe and Karsten Arnbjerg-Nielsen submitted to NHESS**

The authors present a study analyzing the impact of aggregation scale of high resolution DEM, imperviousness and building data on urban pluvial flood risk assessments. The study intends to quantify these impacts and to identify the optimal scale for data aggregation to be used in "flood screening", i.e. for low computational flood hazard and risk assessment considering different flood adaptation scenarios and urban developments. The authors thus deal with a topic that has been of a long standing concern in flood risk research and add an at least useful, but potentially also important contribution to the question of optimal scales to be used in flood risk assessments, here with a particular focus on urban pluvial floods.

The study is generally well designed and presented, the data analysis solid and the conclusion are supported by the results. Overall I don't have any major objections to the presented work, but I suggest to enhance the discussion of the implications of the findings for urban pluvial flood risk assessments in more detail, as well as the generalization/transferability of the results. This would enhance the manuscript and increase the potential impact of the work. In section 5.4 about the limitations of the work the authors state that the regression models likely have to be newly fitted for different topography and urban structures, but that they expect that the identified optimal scales are generic. Unfortunately the authors did not provide any reason why they expect that the optimal scales are generic, i.e. transferable to any other urban flood risk study. This needs to be provided. I actually would challenge this statement. From my experience and understanding of the problem, I would argue that the urban texture/layout also controls the optimal scale for risk assessment. In the context of this work it should control at least the optimal scale of the imperviousness regression. Think of cities with wide roads and sidewalks designed for car traffic (e.g. American suburbs) vs. old towns with narrow streets and sidewalks and/or steep topography (e.g. old European cities with medieval city centers). It can be reasoned that at least the optimal resolution for the imperviousness regression is likely different for these urban structures. If the authors argue against this, proper arguments should be given. Otherwise the limitations of the study results in terms of transferability needs to be extended.

Furthermore, the manuscript would profit if the authors provide recommendation/blueprints of how the presented optimal scales and regressions can be used in other urban flood risk studies/assessments and assessment of flood management/mitigation/urban development plans. What would be the procedure to follow? What are the minimal data and model requirements? This is currently a bit blurry and not well defined. A more detailed illustration of the use of the results/findings would surely increase the uptake of the study in research as well as in practice.

Besides these general concerns, I have some specific small comments listed below.

- The term "flood screening" should be explained/defined in the introduction. The authors expect the reader to be familiar with the term, but this cannot be assumed. Moreover, the term is not widely used (to my knowledge), and thus different readers are likely to associate different meanings to the term.

- I found it occasionally difficult to follow the different aggregation scales used in the different analysis ($\Delta x_{fit}$, $\Delta x_{pred}$). Additionally different terms are used in the manuscript, e.g. $\Delta x_{fit}$ as fitting resolution or data resolution. This should be harmonized. Additionally it would be beneficial to clearly separate these terms in order to easy the understanding of the work done in the different sections, although I also don't have a precise suggestion how this can be achieved. One way could be a clear definition at the start of the method section, e.g. in a table:

| Symbol | Description as used in text | Explanation / used in analysis xy |
|---|---|---|
| $\Delta x_{fit}$ | Data resolution | …. |
| $\Delta x_{pred}$ | Prediction resolution | …. |

The description should then be used constantly throughout the text.

- The regression results are compared to a benchmark simulation based on highly detailed input data. This is totally valid, but ideally a quality statement of the benchmark should be provided. If there is no quality assessment of the benchmark possible (because of lacking data/observations), then there should be at least a statement that benchmark is not validated and could thus also be (far) off reality. Of course this does not touch the validity of the results, because the benchmark could likely be tuned to be close to reality as possible if validation data is available.

- In Figure 3 and associated text it is stated that only 8 aggregation levels (resolutions) for imperviousness (simulated flooded areas) are used for the regression of the damage functions, but there is no reason given for the reduction. I assume that this is because of reduction of possible resolution combinations without compromising the overall results, but it needs to be stated.

- In section 4.1 it is stated that the optimal solution derived from Figure 4 is in the order of 400m, because the curves in Figure 4F have a local minimum at about 400m for prediction resolutions of 500m – 2000m. However, the standard deviation of RMSE for a prediction resolution of 250m has no minimum, but is always below the standard deviation of RMSE of the higher prediction resolution for all fitting resolutions. Therefor I cannot really follow the conclusion that 400m is the optimal fitting resolution for estimating the impervious area. This should be explained better. Moreover, the caption of figure 4 should state that it deals with the regression functions of the imperviousness. This is currently missing, thus impairing the understandability of the figure without reading the associated text section.

- In equation (1) the $a_i$ needs to be explained in the text below. For better understanding the meaning of the equation should be explained in one sentence. The statement "we considered the following relationship" has only a vague relation to the text leaving room for speculation/confusion.

- Page 8, line 156: extend the sentence to "Buildings were not explicitly included in the DEM for flow calculation in this case."

- Section 3.4.1 (page 10, line 215ff): Please provide argument/reasoning for the square root transformation used in equation (6). It is currently unclear why this transformation was performed. Ideally provide a figure in the supplement to justify/explain this transformation. Furthermore the coefficients $b_{xi}$ in equation (6) need to be explained in the text below the equation.

- Page 10, lines 232-234: to improve understandability, clearly state the difference between baseline flood map and the flood maps based on aggregated building data (buildings in the DEM blocking flows and not) again.

- I would feel more comfortable to use the term "coefficient of determination COD" instead of NSE throughout the manuscript. Both have identical meanings, with NSE being adopted in the hydrological modelling community and typically used to compare simulated and observed (discharge) time series, which is clearly not the case in this study. COD is more widely and generically used. However, this is a suggestion, the authors are free to decide.

- Page 11, equation (7): explain subscript "CV 2000". I assume that this refers to "cross validation over the 2000m x 2000m sub-areas", but it needs to be explained.

- Page 17, line 368: what is meant here? "while" does not seem appropriate. Maybe "…, because coarse representations of imperviousness had little effect on the flow dynamics."

- Occasionally the English reads a bit awkward/complicated, which is not of major concern for me, but a grammar check by a native English speaker might improve the manuscript further.

---

## Referee Comment (RC2) · Anonymous Referee #2 · 27 Oct 2019

This paper shows interesting research on the impact of spatial aggregation on urban pluvial flood risk assessments. The article presents a good work, complemented by detailed explanations, tables, and figures. I have some concerns and suggestions.

**Abstract: "Future work needs to focus on training regression approaches where different degrees of flood-awareness in landuse management can be considered". It is not a good practice to provide the future work in the abstract. It is, in turn, expected to be found within the discussion section.

\*\*Method: "Fast urban development models that are useful for exploratory modeling would typically provide outputs resembling those where building areas were rasterized to resolutions between 25 and 500m." Why? Please provide justifications/references.

\*\*Model setup: "To test the impact of spatial data resolution, we fitted regression models to datasets with 80 different resolutions". Did you examine the relationship and ensured that it is a linear relationship? That may lead to a misleading conclusion.

\*\*I would recommend the authors to discuss the transferability of their finding to other places in the discussion section.

\*\*I believe that urban layout setting impacts the flooding according to the findings of some studies (Mustafa et al., 2018). The authors should discuss this point in the discussion section. Mustafa, A., Wei Zhang, X., Aliaga, D.G., Bruwier, M., Nishida, G., Dewals, B., Erpicum, S., Archambeau, P., Pirotton, M., Teller, J., 2018. Procedural generation of flood-sensitive urban layouts. Environ. Plan. B Urban Anal. City Sci. 0, 1–23. https://doi.org/10.1177/2399808318812458

\*\*English needs improvements.

———————————————————

---

## Author Comment (AC1) · 6 Nov 2019

**1 Author's summary**

*We wish to thank the reviewer for taking the time for a thorough review the manuscript and for providing constructive comments. In brief, we agree with the issues raised by the reviewer and will adress them as detailed in our reply to each comment below.*

*This involves the following major changes:*

*• New discussion section "5.5 Generalization and application" which includes a stepwise workflow for deriving suitable scales in a new case study and discusses the limitations linked to topography and urban layout raised by the reviewer*

*• Streamline terminology and symbols related to the different data resolutions in the workflow figure (Fig.3) and throughout the text.*

*• Include scatterplots showing the effect of data transformations in flood damage regression in the supporting material*

*• Improve explanations in the manuscript as suggested by the reviewer in the detailed comments.*

*Both reviewers point out language issues, so we suggest that will have the manuscript proofread by a language editing service before final submission.*

**2  Reviewer Summary**

Review of manuscript "Urban pluvial flood risk assessment - data resolution and spatial scale when developing screening approaches on the micro scale" by Roland Löwe and Karsten Arnbjerg-Nielsen submitted to NHESS The authors present a study analyzing the impact of aggregation scale of high resolution DEM, imperviousness and building data on urban pluvial flood risk assessments. The study intends to quantify these impacts and to identify the optimal scale for data aggregation to be used in "flood screening", i.e. for low computational flood hazard and risk assessment considering different flood adaptation scenarios and urban developments. The authors thus deal with a topic that has been of a long standing concern in flood risk research and add an at least useful, but potentially also important contribution to the question of optimal

scales to be used in flood risk assessments, here with a particular focus on urban pluvial floods.

*This is an appropriate summary of our study.*

**3 Major comments**

The study is generally well designed and presented, the data analysis solid and the conclusion are supported by the results. Overall I don't have any major objections to the presented work, but I suggest to enhance the discussion of the implications of the findings for urban pluvial flood risk assessments in more detail, as well as the generalization/transferability of the results. This would enhance the manuscript and increase the potential impact of the work.

In section 5.4 about the limitations of the work the authors state that the regression models likely have to be newly fitted for different topography and urban structures, but that they expect that the identified optimal scales are generic. Unfortunately the authors did not provide any reason why they expect that the optimal scales are generic, i.e. transferable to any other urban flood risk study. This needs to be provided. I actually would challenge this statement. From my experience and understanding of the problem, I would argue that the urban texture/layout also controls the optimal scale for risk assessment. In the context of this work it should control at least the optimal scale of the imperviousness regression.

Think of cities with wide roads and sidewalks designed for car traffic (e.g. American suburbs) vs. old towns with narrow streets and sidewalks and/or steep topography (e.g. old European cities with medieval city centers). It can be reasoned that at least the optimal resolution for the imperviousness regression is likely different for these urban structures. If the authors argue against this, proper arguments should be given. Otherwise the limitations of the study results in terms of transferability needs to be

extended.

*We agree with the reviewer – optimal scales must depend on the density of urban developments, which can vary between cities. We suggest elaborating on these issues in the discussion section and include them in a new section "5.5 Generalization and application", which will also address the reviewers next comment.*

Furthermore, the manuscript would profit if the authors provide recommendation/blueprints of how the presented optimal scales and regressions can be used in other urban flood risk studies/assessments and assessment of flood management/mitigation/urban development plans. What would be the procedure to follow? What are the minimal data and model requirements? This is currently a bit blurry and not well defined. A more detailed illustration of the use of the results/findings would surely increase the uptake of the study in research as well as in practice.

*The new section "5.5 Generalization and application" will include a stepwise workflow towards creating a screening setup for flood risk in a new case study.*

**4  Detailed comments**

Besides these general concerns, I have some specific small comments listed below.

The term "flood screening" should be explained/defined in the introduction. The authors expect the reader to be familiar with the term, but this cannot be assumed. Moreover, the term is not widely used (to my knowledge), and thus different readers are likely to associate different meanings to the term.

*The term will be defined in the introduction. We also noticed that the term is used in varying ways in the manuscript. We suggest using "Flood screening setup" to refer to the overall setup for fast flood risk assessment (i.e., the combination of fast urban development simulation, simulation of flood hazard and damage calculation), while the*

[Figure]

*simulation of flood hazard should be referred as "fast flood simulation".*

I found it occasionally difficult to follow the different aggregation scales used in the different analysis ($\Delta x_{fit}$, $\Delta x_{pred}$). Additionally different terms are used in the manuscript, e.g. $\Delta x_{fit}$ as fitting resolution or data resolution. This should be harmonized. Additionally it would be beneficial to clearly separate these terms in order to easy the understanding of the work done in the different sections, although I also don't have a precise suggestion how this can be achieved. One way could be a clear definition at the start of the method section, e.g. in a table:

| Symbol | Description as used in text | Explanation used in analysis xy |
|---|---|---|
| $\Delta x_{fit}$ | Data resolution | …. |
| $\Delta x_{pred}$ | Prediction resolution | …. |

The description should then be used constantly throughout the text.

*There are 3 resolutions to distinguish:*

- *$\Delta x_{fit}$ – being the data resolution used when training the regression models (varied from 25 to 2000m - both, in imperviousness and damage regression)*

- *$\Delta x_{pred}$ – being the data resolution at which predictions are generated from the regression models (varied between and 25 and 2000m for imperviousness regression, and kept fixed at 2000m for damage regression)*

- *$\Delta x_b$ – being the resolution of the building data used for predicting imperviousness as input to the 2D flood simulations, as well as to compute the flooding building area as input to the damage regression models*

*We prefer clarifying the usage of different data resolutions in Figure 3 over inserting a new table, because we would expect that the reader would try to understand the*

*dataflow from this figure. The figure will make explicit reference to different data resolutions used in the different parts of the analysis. In addition, we suggest revisiting the text and inserting explicit references to $\Delta x_{fit}$, $\Delta x_{pred}$ and $\Delta x_b$ when discussing resolutions.*

The regression results are compared to a benchmark simulation based on highly detailed input data. This is totally valid, but ideally a quality statement of the benchmark should be provided. If there is no quality assessment of the benchmark possible (because of lacking data/observations), then there should be at least a statement that benchmark is not validated and could thus also be (far) off reality. Of course this does not touch the validity of the results, because the benchmark could likely be tuned to be close to reality as possible if validation data is available.

*We will include a corresponding statement in the Methods section. As reasoned by the reviewer, our aim was to generate flood map which is realistic rather than to reproduce observed conditions.*

In Figure 3 and associated text it is stated that only 8 aggregation levels (resolutions) for imperviousness (simulated flooded areas) are used for the regression of the damage functions, but there is no reason given for the reduction. I assume that this is because of reduction of possible resolution combinations without compromising the overall results, but it needs to be stated. *Indeed, we have performed flood simulations for a limited set of resolutions, because additional simulations require substantial manual effort, provide limited insight and make it difficult to present results in an understandable manner. The statement suggested by the reviewer will be included in the figure caption for Fig. 3.*

In section 4.1 it is stated that the optimal solution derived from Figure 4 is in the order of 400m, because the curves in Figure 4F have a local minimum at about 400m for prediction resolutions of 500m – 2000m. However, the standard deviation of RMSE for a prediction resolution of 250m has no minimum, but is always below the standard

[Figure]

deviation of RMSE of the higher prediction resolution for all fitting resolutions. Therefor I cannot really follow the conclusion that 400m is the optimal fitting resolution for estimating the impervious area. This should be explained better. Moreover, the caption of figure 4 should state that it deals with the regression functions of the imperviousness. This is currently missing, thus impairing the understandability of the figure without reading the associated text section.

*We will clarify the figure caption and provide an explanation for the artefact at 250m prediction resolution in the main text. A detailed explanation is provided below.*

*Detailed explanation:*

*The standard deviation of the estimated regression model coefficients decreases when smaller data resolutions $\Delta x_{fit}$ are considered during model fitting, i.e., we obtain more stable parameter estimates (not shown). The mean parameter estimates approach 1 for very fine data resolutions (not shown), i.e., the regression models only capture the roof area as impervious area. A strong negative bias is thus introduced in the regression predictions of impervious areas.*

*When considering large enough prediction resolutions ($\Delta x_{pred}$), where the pixels containing the buildings also include all the associated impervious areas, this bias leads to strong variability of the RMSE values computed during cross validation, despite smaller variability of the parameter estimates. The variability is driven by different areas being sampled for validation (e.g., more or fewer industrial areas). The bias disappears when coarser data resolutions $\Delta x_{fit}$ are considered (leading also to the increase in COD values in Fig. 4D), however, at the expense of fewer data points being available, leading to instability in the parameter estimates and again an increasing variability of the RMSE values computed during cross validation. The data resolution where $\sigma(RMSE)$ is minimal ($\Delta x_{fit}$ around 400m) indicates the optimal tradeoff, where regression predictions become unbiased, and the data are aggregated only to the necessary level. It is also the resolution where COD values in Fig. 4D reach their maximum.*

*For smaller prediction resolutions ($\Delta x_{pred} = 250m$), we observed an artefact where the biased regression predictions for small data resolutions $\Delta x_{fit}$ do not lead to an increase in $\sigma(RMSE)$. In this case, substantial portions of the impervious areas are located in pixels where building areas are 0. The impervious area predicted by the regression models for these pixels is thus always 0 and does not depend on the regression coefficients. The absolute values of RMSE increase due to the bias. However, the variability of RMSE values (Fig. 4F) becomes determined by how much the predictions of imperviousness close to the buildings vary during cross validation. This variability decreases as the coefficients approach a constant value of 1.*

In equation (1) the $a_i$ needs to be explained in the text below. For better understanding the meaning of the equation should be explained in one sentence. The statement "we considered the following relationship" has only a vague relation to the text leaving room for speculation/confusion.

*Will be adressed*

Page 8, line 156: extend the sentence to "Buildings were not explicitly included in the DEM for flow calculation in this case."

*Will be adressed*

Section 3.4.1 (page 10, line 215ff): Please provide argument/reasoning for the square root transformation used in equation (6). It is currently unclear why this transformation was performed. Ideally provide a figure in the supplement to justify/explain this transformation. Furthermore the coefficients $b_{xi}$ in equation (6) need to be explained in the text below the equation.

*The coefficients will be explained in the paper.*

*Scatterplots showing the relationship between flooded building area and flood damages (with/without data transformation) and a brief explanation will be included in the supporting material. The scatterplots are also attached in the end of this reply. We*

*have, in fact, experimented with a number of power and logarithmic transformations.
The squareroot transformation turned out to be robust and can handle 0 values. The
latter point is a problem particularly with the logarithmic transformation, which amplifies
the impact of outliers and where regression predictions of flood damages for pixels with
a flooded building area of zero are not guaranteed to be zero.*

Page 10, lines 232-234: to improve understandability, clearly state the difference be-
tween baseline flood map and the flood maps based on aggregated building data (build-
ings in the DEM blocking flows and not) again.

*We will include a brief explanation of the baseline flood map in the text.*

I would feel more comfortable to use the term "coefficient of determination COD" in-
stead of NSE throughout the manuscript. Both have identical meanings, with NSE
being adopted in the hydrological modelling community and typically used to compare
simulated and observed (discharge) time series, which is clearly not the case in this
study. COD is more widely and generically used. However, this is a suggestion, the
authors are free to decide.

*We will change "NSE" to "COD" in the text and the result figures.*

Page 11, equation (7): explain subscript "CV 2000". I assume that this refers to "cross
validation over the 2000m x 2000m sub-areas", but it needs to be explained.

*Will be adressed*

Page 17, line 368: what is meant here? "while" does not seem appropriate. Maybe
"..., because coarse representations of imperviousness had little effect on the flow
dynamics."

*Will be rephrased to "Coarse representations of imperviousness and the resulting
change in rainfall-runoff behaviour had little effect in comparison."*

Occasionally the English reads a bit awkward/complicated, which is not of major con-

cern for me, but a grammar check by a native English speaker might improve the manuscript further.

*We suggest having the manuscript checked by a language editing service before final submission.*

[Figure]

**Fig. 1.** Scatterplots of flood damages (Olsen et al., 2015) versus total flooded building area (Delta x_b=200m). Columns: T=20 (left) and 100 years (right). Rows: no transform, sqrt-sqrt, log-log transform

**Fig. 2.** Scatterplots of flood damages (Beckers et al, 2013) versus total flooded building area (Delta x_b=200m). Columns: T=20 (left) and 100 years (right). Rows: no transform, sqrt-sqrt, log-log transform

---

## Author Comment (AC2) · 6 Nov 2019

**1   Author's summary**

*Thank you very much for taking the time to review the manuscript and for providing constructive comments. As outlined below, we have no objections regarding the comments, and hope that our suggested changes will address them appropriately.*

*The following main changes will be implemented following the comments from R2:*

[Figure]

*• New discussion section 5.5 ("Generalization and application"), which focuses on issues of generalization and transferability.*

*• Rephrasing the beginning of the Methods section to explain what outputs can be expected from urban development models and how this links to the building data applied in our work.*

*Both reviewers point out language issues, so we suggest that we will have the manuscript proofread by a language editing service before final submission.*

**2  Reviewer summary**

This paper shows interesting research on the impact of spatial aggregation on urban pluvial flood risk assessments. The article presents a good work, complemented by detailed explanations, tables, and figures.

*Thank you for the positive feedback.*

**3  Reviewer comments**

I have some concerns and suggestions. 1. Abstract: "Future work needs to focus on training regression approaches where different degrees of flood-awareness in landuse management can be considered". It is not a good practice to provide the future work in the abstract. It is, in turn, expected to be found within the discussion section.

*We will remove the sentence from the abstract. Issues related to the application of our approach and required future work will be summarized in a new Section 5.5 ("Generalization and application"), following a similar comment from R1.*

2. Method: "Fast urban development models that are useful for exploratory modeling would typically provide outputs resembling those where building areas were rasterized to resolutions between 25 and 500m." Why? Please provide justifications/references.

*We suggest to reformulate the paragraph as illustrated below:*

*"Hydrological modeling and flood damage assessment are commonly performed based on polygon data characterizing the urban layout. Fast, raster-based urban development models instead provide information about the building area inside a pixel, or the land use mix inside a pixel. The latter can, through an assumed building density, be translated into building areas. Typically, these models operate with raster resolutions in the order of 100 to 200m (Bach et al.,2018; Mustafa et al., 2018; Fuglsang et al., 2013). Such coarse input data will affect both rainfall runoff simulations, i.e., the location where flood hazards occur, and are likely to be incompatible with flood damage assessments derived for polygon data. To analyze issues arising in different parts of the pluvial flood risk modeling chain, we performed hydrological assessments considering imaginary urban development model outputs in the form of rasterized building data with resolutions between 25 and 2000m.. . .."*

3. Model setup: "To test the impact of spatial data resolution, we fitted regression models to datasets with 80 different resolutions". Did you examine the relationship and ensured that it is a linear relationship? That may lead to a misleading conclusion.

*We did. Scatterplots of building area vs. impervious area had already been included in the supporting information and suggest linear behaviour (Figure S1). However, we suggest reformulating the sentence under the equation to clarify that these plots are provided:*

*"Scatterplots of impervious area versus building area were included in the supporting information (Figure S1). We have not included an intercept in Eq. (1) to ensure undeveloped areas are assigned an imperviousness of 0, and because the scatterplots did not suggest that an intercept would be necessary. For fine data resolutions this leads*

*to biased regression predictions."*

*Reviewer 1 had a similar comment regarding the data transformation applied in damage regression. We refer to page C8 in our reply to reviewer 1 ("'Section 3.4.1 (page 10, line 215ff)'")*

4. I would recommend the authors to discuss the transferability of their finding to other places in the discussion section.

*Following your comment and similar comments from reviewer 1, we suggest including a new section "5.5 Generalization and application" in the manuscript which discusses these issues.*

5. I believe that urban layout setting impacts the flooding according to the findings of some studies (Mustafa et al., 2018). The authors should discuss this point in the discussion section. Mustafa, A., Wei Zhang, X., Aliaga, D.G., Bruwier, M., Nishida, G., Dewals, B., Erpicum, S., Archambeau, P., Pirotton, M., Teller, J., 2018. Procedural generation of flood-sensitive urban layouts. Environ. Plan. B Urban Anal. City Sci. 0, 1–23. https://doi.org/10.1177/2399808318812458

*This point will be included in the new section "5.5 Generalization and application". We prefer to refer to the companion paper from the same group, which explicitly assesses the impact of different characteristics of urban layouts on flood hazard.*

*Bruwier, M., Mustafa, A., Aliaga, D. G., Archambeau, P., Erpicum, S., Nishida, G., ... Dewals, B. (2018). Influence of urban pattern on inundation flow in floodplains of lowland rivers. Science of the Total Environment, 622–623, 446–458. https://doi.org/10.1016/j.scitotenv.2017.11.325*

6. English needs improvements.

*We will have the manuscript checked by a language editing service before final submission.*

---

## Author Response (AR1)

**Natural Hazards and Earth System Sciences (NHESS) Journal**

Att. Prof. Thomas Glade

**NHESS-2019-272 Revision 1**

Dear Prof. Glade,

We wish to thank you for considering our manuscript.

18 December 2019
ROLO

We agree with all points raised by the reviewers and made the following changes:

- Included a new discussion section 5.5 which focuses on the potential for generalization of our approach and suggests a course of action for implementing our findings in new case studies
- Included a new section S6 in the supporting information which elaborates the choice of data transformation in flood damage regression
- Clarified terminology on data resolutions / fitting resolutions / prediction resolutions in the workflow figure (Fig. 3) and the main text
- Clarified explanations throughout the manuscript as outlined in our replies to the reviewers detailed comments.

Both reviewers commented on minor language issues. We confirmed with Copernicus that the journal performs language copy-editing and therefore suggest that this is addressed after acceptance.

We hope to have addressed the comments appropriately and look forward to your reply.

**Best regards,**

**Roland Löwe and Karsten Arnbjerg-Nielsen**

Included below

- Point by point reply to reviewers and indication of where changes were made in the manuscript
- Manuscript version with changes to text and tables highlighted

[Figure]

REG-no. DK 30 06 09 46

**DTU Environment**
Department of
Environmental Engineering

Bygningstorvet
Building 115
2800 Kgs. Lyngby
Denmark

Tel.. +45 45 25 16 00
Dir. +45 45251694
Fax +45 45 93 28 50

rolo@env.dtu.dk
www.env.dtu.dk

*We wish to thank the reviewer for taking the time for a thorough review the manuscript and for providing constructive comments. In brief, we agree with the issues raised by the reviewer and will adress them as detailed in our reply to each comment below.*

*This involves the following major changes:*
- *New discussion section "5.5 Generalization and application" which includes a stepwise workflow for deriving suitable scales in a new case study and discusses the limitations linked to topography and urban layout raised by the reviewer*
- *Streamline terminology and symbols related to the different data resolutions in the workflow figure (Fig.3) and throughout the text.*
- *Include scatterplots showing the effect of data transformations in flood damage regression in the supporting material*
- *Improve explanations in the manuscript as suggested by the reviewer in the detailed comments.*

*Both reviewers point out language issues, so we suggest that will have the manuscript proofread by a language editing service before final submission.*

**Review of manuscript „Urban pluvial flood risk assessment - data resolution and spatial scale when developing screening approaches on the micro scale" by Roland Löwe and Karsten Arnbjerg-Nielsen submitted to NHESS**

The authors present a study analyzing the impact of aggregation scale of high resolution DEM, imperviousness and building data on urban pluvial flood risk assessments. The study intends to quantify these impacts and to identify the optimal scale for data aggregation to be used in "flood screening", i.e. for low computational flood hazard and risk assessment considering different flood adaptation scenarios and urban developments. The authors thus deal with a topic that has been of a long standing concern in flood risk research and add an at least useful, but potentially also important contribution to the question of optimal scales to be used in flood risk assessments, here with a particular focus on urban pluvial floods.

*This is an appropriate summary of our study.*

The study is generally well designed and presented, the data analysis solid and the conclusion are supported by the results. Overall I don't have any major objections to the presented work, but I suggest to enhance the discussion of the implications of the findings for urban pluvial flood risk assessments in more detail, as well as the generalization/transferability of the results. This would enhance the manuscript and increase the potential impact of the work. In section 5.4 about the limitations of the work the authors state that the regression models likely have to be newly fitted for different topography and urban structures, but that they expect that the identified optimal scales are generic. Unfortunately the authors did not provide any reason why they expect that the optimal scales are generic, i.e. transferable to any other urban flood risk study. This needs to be provided. I actually would challenge this statement. From my experience and understanding of the problem, I would argue that the urban texture/layout also controls the optimal scale for risk assessment. In the context of this work it should control at least the optimal scale of the imperviousness regression.
Think of cities with wide roads and sidewalks designed for car traffic (e.g. American suburbs) vs. old towns with narrow streets and sidewalks and/or steep topography (e.g. old European cities with medieval city centers). It can be reasoned that at least the optimal resolution for the

imperviousness regression is likely different for these urban structures. If the authors argue against this, proper arguments should be given. Otherwise the limitations of the study results in terms of transferability needs to be extended.

*We agree with the reviewer – optimal scales must depend on the density of urban developments, which can vary between cities. We have elaborated on these issues in the discussion section and use them to introduce the application workflow in the new section "5.5 Generalization and application" (l434ff)*

Furthermore, the manuscript would profit if the authors provide recommendation/blueprints of how the presented optimal scales and regressions can be used in other urban flood risk studies/assessments and assessment of flood management/mitigation/urban development plans. What would be the procedure to follow? What are the minimal data and model requirements? This is currently a bit blurry and not well defined. A more detailed illustration of the use of the results/findings would surely increase the uptake of the study in research as well as in practice.

*The new section "5.5 Generalization and application" now includes a stepwise workflow towards creating a screening setup for flood risk in a new case study.*

**Besides these general concerns, I have some specific small comments listed below.**
The term "flood screening" should be explained/defined in the introduction. The authors expect the reader to be familiar with the term, but this cannot be assumed. Moreover, the term is not widely used (to my knowledge), and thus different readers are likely to associate different meanings to the term.

*The term is now defined in the introduction (l30). We also noticed that the term was used in varying ways in the original text. "Flood screening setup" does now refer to the overall setup for fast flood risk assessment (i.e., the combination of fast urban development simulation, simulation of flood hazard and damage calculation), while the simulation of flood hazard is referred to as "fast flood simulation".*

- I found it occasionally difficult to follow the different aggregation scales used in the different analysis ($\Delta x_{fit}$, $\Delta x_{pred}$). Additionally different terms are used in the manuscript, e.g. $\Delta x_{fit}$ as fitting resolution or data resolution. This should be harmonized. Additionally it would be beneficial to clearly separate these terms in order to easy the understanding of the work done in the different sections, although I also don't have a precise suggestion how this can be achieved. One way could be a clear definition at the start of the method section, e.g. in a table:

| Symbol | Description as used in text | Explanation / used in analysis xy |
|---|---|---|
| $\Delta x_{fit}$ | Data resolution | …. |
| $\Delta x_{pred}$ | Prediction resolution | …. |

The description should then be used constantly throughout the text.
*There are 3 resolutions to distinguish:*
- *$\Delta x_{fit}$ – being the data resolution used when training the regression models (varied from 25 to 2000m - both, in imperviousness and damage regression)*
- *$\Delta x_{pred}$ – being the data resolution at which predictions are generated from the regression models (varied between and 25 and 2000m for imperviousness regression, and kept fixed at 2000m for damage regression)*
- *$\Delta x_b$ – being the resolution of the building data used for predicting imperviousness as input to the 2D flood simulations, as well as to compute the flooding building area as input to the damage regression models*

*We preferred clarifying the usage of different data resolutions in Figure 3 over inserting a new*

*table, because we would expect that the reader would try to understand the dataflow from this figure. The figure now makes explicit reference to different data resolutions used in the different parts of the analysis. In addition, we have revisited the text. We inserted explicit references to $\Delta x_{fit}$, $\Delta x_{pred}$ and $\Delta x_b$ when discussion resolutions and removed the term "fitting resolutions".*

- The regression results are compared to a benchmark simulation based on highly detailed input data. This is totally valid, but ideally a quality statement of the benchmark should be provided. If there is no quality assessment of the benchmark possible (because of lacking data/observations), then there should be at least a statement that benchmark is not validated and could thus also be (far) off reality. Of course this does not touch the validity of the results, because the benchmark could likely be tuned to be close to reality as possible if validation data is available.

*We have included a corresponding statement in the Methods section (l176). As reasoned by the reviewer, our aim was to generate flood map which is realistic rather than to reproduce observed conditions.*

In Figure 3 and associated text it is stated that only 8 aggregation levels (resolutions) for imperviousness (simulated flooded areas) are used for the regression of the damage functions, but there is no reason given for the reduction. I assume that this is because of reduction of possible resolution combinations without compromising the overall results, but it needs to be stated.

*Indeed, we have performed flood simulations for a limited set of resolutions, because additional simulations require substantial manual effort, provide limited insight and make it difficult to present results in an understandable manner.*
*The statement suggested by the reviewer is now included in the figure caption for Fig. 3.*

In section 4.1 it is stated that the optimal solution derived from Figure 4 is in the order of 400m, because the curves in Figure 4F have a local minimum at about 400m for prediction resolutions of 500m – 2000m. However, the standard deviation of RMSE for a prediction resolution of 250m has no minimum, but is always below the standard deviation of RMSE of the higher prediction resolution for all fitting resolutions. Therefor I cannot really follow the conclusion that 400m is the optimal fitting resolution for estimating the impervious area. This should be explained better. Moreover, the caption of figure 4 should state that it deals with the regression functions of the imperviousness. This is currently missing, thus impairing the understandability of the figure without reading the associated text section.

*We have clarified the figure caption and provide an explanation for the artefact at 250m prediction resolution in the main text. A detailed explanation is provided below, but it should hopefully also be clear from the paper now (l273-293).*
*-----------------------------------------------------------------------------------------------------------------------------*
*The standard deviation of the estimated regression model coefficients decreases the smaller data resolutions $\Delta x_{fit}$ are considered during model fitting, i.e., we obtain to more stable parameter estimates (not shown). The mean parameter estimates approach 1 the smaller the data resolution becomes (not shown), i.e., the regression models only capture the roof area as impervious area. A strong negative bias is thus introduced in the regression predictions of impervious areas.*

*When considering large enough prediction resolutions ($\Delta x_{pred}$), where the pixels containing the buildings also include all the associated impervious areas, this bias leads to strong variability of the RMSE values computed during cross validation, despite smaller variability of the parameter estimates. The variability is driven by different areas being sampled for validation (e.g., more or fewer industrial areas). The bias disappears when coarser data resolutions $\Delta x_{fit}$ are considered*

*(leading also the increase in COD values in Fig. 4D), however, at the expense of fewer data points being available, leading to instability in the parameter estimates and again an increasing variability of the RMSE values computed during cross validation. The data resolution where $\sigma(RMSE)$ is minimal ($\Delta x_{fit} \sim 400m$) indicates the optimal tradeoff, where regression predictions become unbiased and the data are aggregated only to the necessary level. It is also the resolution where COD values in Fig. 4D reach their maximum.*

*For smaller prediction resolutions ($\Delta x_{pred} = 250m$), we observed an artefact where the biased regression predictions for small data resolutions $\Delta x_{fit}$ do not lead to an increase in $\sigma(RMSE)$. In this case, substantial portions of the impervious areas are located in pixels where building areas are 0. The impervious area predicted by the regression models for these pixels is thus always 0 and does not depend on the regression coefficients. The absolute values of RMSE increase due to the bias. However, the variability of RMSE values (Fig. 4F) becomes determined by how much the predictions of imperviousness close to the buildings vary during cross validation. This variability decreases as the coefficients approach a constant value of 1.*

- In equation (1) the $a_i$ needs to be explained in the text below. For better understanding the meaning of the equation should be explained in one sentence. The statement "we considered the following relationship" has only a vague relation to the text leaving room for speculation/confusion.

*Fixed (l125)*

- Page 8, line 156: extend the sentence to "Buildings were not explicitly included in the DEM for flow calculation in this case."

*Fixed, we now also refer to flow calculation in the other 2 bullets (l154).*

- Section 3.4.1 (page 10, line 215ff): Please provide argument/reasoning for the square root transformation used in equation (6). It is currently unclear why this transformation was performed. Ideally provide a figure in the supplement to justify/explain this transformation. Furthermore the coefficients $b_{xi}$ in equation (6) need to be explained in the text below the equation.

*The coefficients are now explained in the paper (l226)*

*We have included scatterplots and a brief reasoning in the supporting material (Figures S4 and S5). We have, in fact, experimented with a number of power and logarithmic transformations. The squareroot transformation turned out to be robust and can handle 0 values. The latter point is a problem particularly with the logarithmic transformation, which amplifies the impact of outliers (see Figures S4 and S5) and where regression predictions of flood damages for pixels with a flooded building area of zero are not guaranteed to be zero.*

- Page 10, lines 232-234: to improve understandability, clearly state the difference between baseline flood map and the flood maps based on aggregated building data (buildings in the DEM blocking flows and not) again.

*We have included a brief explanation of the baseline flood map in the text (l239).*

- I would feel more comfortable to use the term "coefficient of determination COD" instead of NSE throughout the manuscript. Both have identical meanings, with NSE being adopted in the hydrological modelling community and typically used to compare simulated and observed

(discharge) time series, which is clearly not the case in this study. COD is more widely and generically used.

However, this is a suggestion, the authors are free to decide.

*We have changed "NSE" to "COD" in the text and the result figures.*

- Page 11, equation (7): explain subscript "CV 2000". I assume that this refers to "cross validation over the 2000m x 2000m sub-areas", but it needs to be explained.

*Fixed (l256)*

- Page 17, line 368: what is meant here? "while" does not seem appropriate. Maybe "…, because coarse representations of imperviousness had little effect on the flow dynamics."

*Rephrased to "Coarse representations of imperviousness and the resulting change in rainfall-runoff behaviour had little effect in comparison." (l384)*

- Occasionally the English reads a bit awkward/complicated, which is not of major concern for me, but a grammar check by a native English speaker might improve the manuscript further.

*We noticed that the journal performs language copy-editing and therefore suggest that this is addressed after acceptance.*

*Thank you very much for taking the time to review the manuscript and for providing constructive comments. As outlined below, we have no objections regarding the comments, and we hope to have addressed them appropriately.*

This paper shows interesting research on the impact of spatial aggregation on urban pluvial flood risk assessments. The article presents a good work, complemented by detailed explanations, tables, and figures. I have some concerns and suggestions.

1. Abstract: "Future work needs to focus on training regression approaches where different degrees of flood-awareness in landuse management can be considered". It is not a good practice to provide the future work in the abstract. It is, in turn, expected to be found within the discussion section.

   *The sentence was removed from the abstract. Issues related to the application of our approach and required future work is now summarize in Section 5.5 ("Generalization and application") which was introduced following a similar comment from R1 (l435ff).*

2. Method: "Fast urban development models that are useful for exploratory modeling would typically provide outputs resembling those where building areas were rasterized to resolutions between 25 and 500m." Why? Please provide justifications/references.

   *We reformulated the paragraph as illustrated below to clarify the context (l 83ff):*
   *"Hydrological modeling and flood damage assessment are commonly performed based on polygon data characterizing the urban layout. Fast, raster-based urban development models instead provide information about the building area inside a pixel or the land use mix inside a pixel, which, through an assumed building density, can be translated into building areas. Typically, these models operate with raster resolutions in the order of 100 to 200m (Bach et al.,2018; Mustafa et al., 2018; Fuglsang et al., 2013). Such coarse input data will affect both rainfall runoff simulations, i.e., the location where flood hazards occur, and are likely to be incompatible with flood damage assessments derived for polygon data. To analyze issues arising in different parts of the pluvial flood risk modeling chain, we performed hydrological assessments considering imaginary urban development model outputs in the form of rasterized building data with resolutions between 25 and 2000m....."*

3. Model setup: "To test the impact of spatial data resolution, we fitted regression models to datasets with 80 different resolutions". Did you examine the relationship and ensured that it is a linear relationship? That may lead to a misleading conclusion.

   *We did. Scatterplots of building area vs. impervious area had already been included in the supporting information (Figure S1). However, we have reformulated the sentence under the equation to clarify that these plots are provided (l128):*
   *"Scatterplots of impervious area versus building area were included in the supporting information (Figure S1). We have not included an intercept in Eq. (1) to ensure undeveloped areas are assigned an imperviousness of 0, and because the scatterplots did not suggest that an intercept would be necessary. For fine data resolutions this leads to biased regression predictions."*
   *Reviewer 1 had a similar comment regarding the data transformation applied in damage regression. We refer to page C8 in our reply to reviewer 1 ("'Section 3.4.1 (page 10, line 215ff)'").*

4. I would recommend the authors to discuss the transferability of their finding to other places in the discussion section.

*Following your comment and similar comments from reviewer 1, we have included a new section "5.5 Generalization and application" in the manuscript (l434ff).*

5. I believe that urban layout setting impacts the flooding according to the findings of some studies (Mustafa et al., 2018). The authors should discuss this point in the discussion section. Mustafa, A., Wei Zhang, X., Aliaga, D.G., Bruwier, M., Nishida, G., Dewals, B., Erpicum, S., Archambeau, P., Pirotton, M., Teller, J., 2018. Procedural generation of flood-sensitive urban layouts. Environ. Plan. B Urban Anal. City Sci. 0, 1–23. https://doi.org/10.1177/2399808318812458

   *This point is now included in the new section "5.5 Generalization and application" (l441). We prefer to refer to the companion paper from the same group, which explicitly assesses the impact of different characteristics of urban layouts on flood hazard.*
   *Bruwier, M., Mustafa, A., Aliaga, D. G., Archambeau, P., Erpicum, S., Nishida, G., … Dewals, B. (2018). Influence of urban pattern on inundation flow in floodplains of lowland rivers. Science of the Total Environment, 622–623, 446–458. https://doi.org/10.1016/j.scitotenv.2017.11.325*

6. English needs improvements.

   *We noticed that the journal performs language copy-editing and therefore suggest that this is adressed after acceptance.*

[revised manuscript text omitted]

$$A_{imp,j} = \sum_i a_i \cdot A_{bf,i,j} \tag{1}$$

 Scatterplots of impervious area versus building area were included in the supporting information (Figure S1). We have not included an intercept in Eq. (1) to ensure undeveloped areas are assigned an imperviousness of 0, and because  the scatterplots did not suggest that an intercept would be necessary.

[Figure]

**Figure 3.** Outline of the analysis steps performed in this paper. Letters A to D refer to the part of the methods sections were the corresponding step is detailed. The dashed line illustrates the case where flood maps from the baseline simulation were used to derive flooded building areas as input to damage regression. Note that the second baseline 2D flood simulation where buildings were not inserted in the DEM is not shown in the flow chart. Steps B and D were performed for a set of 8 selected building raster resolutions $\Delta x_b$ to reduce the number of possible resolution combinations without compromising the overall results.

For fine data resolutions this leads to biased regression predictions. While the dataset certainly is subject to spatial autocorrelation, the regression models provided strong predictive performance and we have therefore not investigated the matter further.

To test the impact of spatial data resolution, we fitted regression models to datasets with 80 different resolutions $\Delta x_{fit}$, ranging from 25 to 2000m in steps of 25m. The regression coefficients identified for each resolution were then used to predict
145 imperviousness at 80 different aggregation levels $\Delta x_{pred}$, ranging from 25 to 2000m. We embedded our tests into a cross-validation setup where 80% of the dataset were used for calibration and 20% for model validation. If $\Delta x_{pred} > \Delta x_{fit}$ we sampled from the pixels of the dataset used for prediction, and otherwise from the pixels of the fitting dataset. For cross-validation, a pixel from the dataset with finer resolution was linked to the pixel of the dataset with coarser resolution with which it shared the greatest overlap. The cross-validation procedure was repeated $k = 1000$ times, i.e., a total of $80 \cdot 80 \cdot 1000$
150 regression models was considered.

**3.1.2 Performance assessment**

During each iteration, we computed  bias ratio $RBIAS_{Aimp,k}$  :

$$RMSE_{Aimp,k} = \sqrt{1/n \cdot \sum_{j} (A_{imp,pred,j} - A_{imp,obs,j})^2} \tag{2}$$

$$\underline{NSE}COD_{Aimp,k} = 1 - \frac{\sum_{j} (A_{imp,pred,j} - A_{imp,obs,j})^2}{\sum_{j} (A_{obs,pred,j} - \overline{A_{imp,obs}})^2} \frac{\sum_{j} (A_{imp,pred,j} - A_{imp,obs,j})^2}{\sum_{j} (A_{imp,obs,j} - \overline{A_{imp,obs}})^2} \tag{3}$$

$$RBIAS_{Aimp,k} = \sum_{j} A_{imp,pred,j} / \sum_{j} A_{imp,obs,j}, \tag{4}$$

where $A_{imp,pred,j}$ and $A_{imp,obs,j}$ were predicted and observed impervious areas for a pixel $j$ in the validation dataset and $\overline{A_{imp,obs}}$ was the average imperviousness of all pixels $j$ in the validation dataset. We considered the median of $RBIAS_{Aimp,k}$ and  $COD_{Aimp,k}$ over all $k$ iterations as measures of goodness of fit, and the standard deviation $\sigma(RMSE_k)$ of $RMSE_{Aimp,k}$ as a measure of how reliably the model could be identified for a given combination of $\Delta x_{fit}$ and $\Delta x_{pred}$.

**3.2 B - 2D flood simulations**

**3.2.1 Model setup**

We performed 2D flood simulations of pluvial hazards for ten different models, considering:

- a model where imperviousness was determined from the original imperviousness dataset, and where buildings were included in the DEM for flow calculation (baseline model),

- a model where imperviousness was determined from the original imperviousness dataset, and where buildings were not included in the DEM for flow calculation (baseline without buildings), and

- models where imperviousness was derived considering the regression relationship shown in the supporting material (Sect. S2), and considering building data aggregated to resolutions $\Delta x_b$ of 25, 50, 100, 200, 300, 500, 750 and 1000m as input. Buildings were not explicitly included in the DEM for flow calculation in this case.

Our 2D modelling approach was the exact same as used by Kaspersen et al. (2017) for the same case study area. The 2D surface flow model was implemented in MIKE 21 (DHI, 2016) using a grid size of 5m. Simulations were performed for Chicago design storms (CDS) with return periods of 20 and 100 years and durations of 4 hours. Rainfall-runoff computations were performed for each grid cell during each time step of a simulated event, and the runoff created in each cell was then included in the simulation of surface water flows.

As in Kaspersen et al. (2017), runoff $R_t$ in time step $t$ for each 5m pixel was computed as

$$R_t = P_t - f_t(1 - IS) - P_{t,RP5}IS, \tag{5}$$

where $P_t$ was the rain intensity and $IS$ the ratio of impervious area in a pixel to its total area. The effective infiltration intensity $f_t(1-IS)$ in a cell was computed based on a constant infiltration rate $f_t = 29.3mm \cdot h^{-1}$. On the impervious portions of a pixel, the rain intensity $P_{t,RP5}$ of a 5 year design storm at the same time step $t$ was subtracted from the rain intensity to simulate the effect of drainage systems.

Impervious areas linked to major roads (Figure 1) were preserved throughout all simulations. In an urban development simulation, main roads would need to be considered explicitly, instead of being lumped into a regression prediction of imperviousness with building areas as the only input. As an example, we included maps of infiltration rates $f_t(1-IS)$ derived for two building datasets in the supporting information, Sect. S3.

The 2D flood model was not calibrated to reflect observed flooding in the catchment. While the simulated flood maps may not coincide with reality, they provide a realistic baseline for the further analysis.

[revised manuscript text omitted]

$$\underline{NSE}COD_{D,CV2000,k} = 1 - \frac{\sum_j (D_{pred,j,k} - D_{baseline,j})^2}{\sum_j (D_{baseline,j} - \overline{D_{baseline}})^2} \tag{7}$$

The index $CV2000$ indicates that cross validation was performed on a spatial scale of 2000m. In addition, we computed the total damage ratio $DR_{tot,k}$ considering all subareas $j$ in the validation dataset as

$$DR_{tot,k} = \sum_j D_{pred,j,k} / \sum_j D_{baseline,j}. \tag{8}$$

Median values of $\sim\!\!NSE_{D,CV2000,k}$ $COD_{D,CV2000,k}$ and $DR_{tot,k}$ were considered in the analysis of results. For the cases where flooded building areas $A_{flooded,WL[i]}$ were derived based on the flood map from the baseline simulation, scores were marked with subscript **BF**.

**4  Results**

The results section was structured into the same parts that were also highlighted in Fig. (3). Performance scores related to the simulation of flood hazards and the assessment of flood damages (parts B to D) were collected in Tables 2 and 3, distinguishing results for building data with varying resolutions $\Delta x_b$.

**4.1  A - Estimation of impervious areas**

Figure 4 summarizes $\sim\!\!NSE_{Aimp,k}$ $COD_{Aimp,k}$, $RBIAS_{Aimp,k}$ and $RMSE_{Aimp,k}$ where regression models for impervious area were fitted for varying data resolutions ($\Delta x_{fit}$), and where the coefficients fitted for one resolution were used to predict impervious areas considering building data aggregated to varying resolutions as input ($\Delta x_{pred}$). Subfigures A-C show histograms of the score values obtained during 1000 cross validation iterations for the combination $\Delta x_{fit} = \Delta x_{pred} = 500m$, while subfigures D-F show median values of $\sim\!\!NSE_{Aimp,k}$ $COD_{Aimp,k}$ and $RBIAS_{Aimp,k}$ and the standard deviation of $RMSE_{Aimp,k}$ obtained for each of the $80 \cdot 80$ combinations of $\Delta x_{fit}$ and $\Delta x_{pred}$.

When the regression models were fitted to data with resolutions below approximately 250m, the relationship between building footprint areas and imperviousness could not be identified, because building footprint areas would then not necessarily be located in the same pixels as the associated features of the urban layout (e.g., sidewalks). Regression coefficients approached 1 for the finest data resolutions $\Delta x_{fit}$ and hardly varied during cross validation (not shown). This lead to low values for $\sim\!\!NSE_{Aimp}$ $COD_{Aimp}$ and an under-prediction of the total imperviousness ($RBIAS_{Aimp} < 1$).  Considering the prediction resolution $\Delta x_{pred}$, values of $COD_{Aimp}$ above 0.95 were achieved  at spatial scales above 500m. For finer spatial scales, there would be random variations in the imperviousness that could not be explained by the amount of building footprint areas alone (see also Fig. (S1)).

While the median predictive performance of the regression models ($\sim\!\!NSE_{Aimp}$ $COD_{Aimp}$ and $RBIAS_{Aimp}$) remained constant for  data resolutions $\Delta x_{fit}$ between approximately 250 and 2000m, the standard deviation of the $RMSE$ values obtained for a fixed prediction resolution was minimal for  data resolutions in the order of 400m  (Fig. (4F)). For coarser data resolutions there would be a larger portion of the cross validation iterations where the regression models would not be properly identified. This behavior was considered plausible, because coarser  data resolutions are accompanied by a loss of information on spatial variability, and because the  number of data points  decreases. For finer data resolutions $\Delta x_{fit}$, the negative bias in predicted imperviousness similarly lead to an increase of $\sigma(RMSE_k)$, because prediction errors varied depending on which areas were sampled for validation. This effect was not observed for $\Delta x_{pred} = 250m$, because the impervious areas that were

[Figure]

**Figure 4.** Results for regression models for impervious areas. Subfigures A-C: Histograms of $\cancel{NSE_{Aimp,k}}COD_{Aimp,k}$, $RBIAS_{Aimp,k}$ and $RMSE_k$ obtained during 1000 cross validation iterations $k$ for the combination of fitting resolution $\Delta x_{fit} = 500m$ and prediction resolution $\Delta x_{pred} = 500m$. Red lines in subfigures A and B indicate median values. Subfigures D and E: median values of $\cancel{NSE_{Aimp,k}}COD_{Aimp,k}$ and $RBIAS_{Aimp,k}$ obtained for varying combinations of $\Delta x_{fit}$ and $\Delta x_{pred}$. Subfigures F: standard deviation (log-transformed) of $RMSE_k$ obtained for varying values of $\Delta x_{fit}$ and selected values of $\Delta x_{pred}$ (dots with varying colours).

not captured during parameter estimation were then also in the validation phase largely located in pixels where the building area was zero (leading to a predicted imperviousness of constant zero).

[revised manuscript text omitted]
 $\cancel{NSE_{D,CV2000}}$ $COD_{D,CV2000}$ varied when different data resolutions $\Delta x_{fit}$ were applied for regression model fitting$\cancel{(\Delta x_{fit})}$, and when different building data resolutions $\Delta x_b$ were considered for both parametrizing imperviousness in the surface flood simulations and for computing flooded building area as input to the regression models. As the computed score values were noisy (see Sect. 3.4), we have displayed smoothed lines (R function "'loess'" with parameter

[revised manuscript text omitted]

Our 2D flood modeling approach was a simplified representation of the urban water cycle. This approach was justified as our intention was to evaluate which spatial scales should be considered in the development of flood screening approaches. For detailed assessment of the risk we would recommend 1D-2D calculation methods to more accurately represent where flooding occurs in the catchment.

**5.5 Generalization and application**

The regression parameters for imperviousness are likely to depend on topography and urban layout (e.g., degree of urban creep and density of the urban developments). In addition, the optimal data resolution for identifying regression relationships is likely to depend on the urban layout, with coarser data resolutions being optimal in less densely developed cities. This implies that regression models can be transferred between cities with similar urban layout and topography, but in many cases it will be necessary to identify optimal spatial scales and model parameters for the specific case study.

For flood damage regression, optimal spatial scales and the identified regression models additionally depend on the approach which is used for calculating flood damages. Further, the level of damages incurred by a given amount of flooded area must be expected to depend on the location of sinks and flow paths in the specific case study, and the degree to which urban planning was performed in a flood aware manner (Bruwier et al., 2018). We thus expect that these regression models always have to be identified for the specific case study. Considering the impact of different approaches to landuse planning is an important line of future research in the development of flood screening approaches.  This effect can be considered by training regression models to different datasets

Based on the considerations above, we suggest the following work flow for developing a fast flood risk screening setup in a new case study:

1. Obtain vector based building data and highly resolved imperviousness data from aerial imagery as base data characterizing the urban layout.

2. Perform hydrodynamic flood simulations (e.g., 1D-2D) for the case study to derive a baseline flood map and compute flood damages.

3. Train regression models for impervious area and identify a suitable data resolution $\Delta x_{fit}$ using cross-validation as demonstrated in this paper (see computer code in Löwe (2019)).

4. Use the predicted impervious area as input to fast flood simulation tools (e.g., Jamali et al. (2019)) and generate flood map.

5. Use the flood map and rasterized building data to train damage regression models. Identify suitable resolutions for training data ($\Delta x_{fit}$) and building rasters ($\Delta x_b$) using cross-validation.

6. Apply setup - simulate urban development in raster format, predict impervious area based on the simulated building areas, use predicted imperviousness for rainfall-runoff calculation in fast flood simulation tool and compute flood damages based on the generated flood map, simulated building areas and damage regression model.

**6 Conclusions**

[revised manuscript text omitted]

[1]$COD$ was computed by aggregating damages derived for the baseline simulation without buildings to 2000m, and comparing these results against the baseline simulation with buildings. $DR$ was computed by computing the ratio of total damages in both simulations.